



# Development of a cloud particle sensor for radiosonde sounding

Masatomo Fujiwara[1,2], Takuji Sugidachi[1,*], Toru Arai[1,**], Kensaku Shimizu[3], Mayumi Hayashi[4], Yasuhisa Noma[4], Hideaki Kawagita[4], Kazuo Sagara[4], Taro Nakagawa[5], Satoshi Okumura[5], Yoichi Inai[2], Takashi Shibata[6], and Suginori Iwasaki[7]

[1]Graduate School of Environmental Science, Hokkaido University, Sapporo, 060-0810, Japan
[2]Faculty of Environmental Earth Science, Hokkaido University, Sapporo, 060-0810, Japan
[3]Meisei Electric Co., Ltd., Isesaki, 372-8585, Japan
[4]Shinyei Technology Co., Ltd., Kobe, 650-0047, Japan
[5]Shinyei Kaisha, Kobe, 651-0178, Japan
[6]Graduate School of Environmental Studies, Nagoya University, Nagoya, 464-8601, Japan
[7]Department of Earth and Ocean Sciences, National Defense Academy, Yokosuka, 239-8686, Japan
[*]Now at: Meisei Electric Co., Ltd., Isesaki, 372-8585, Japan
[**]Now at: MTG Co., Ltd., Nagoya, 453-0041, Japan

*Correspondence to:* Masatomo Fujiwara (fuji@ees.hokudai.ac.jp)

**Abstract.** A meteorological balloon-borne cloud sensor called the Cloud Particle Sensor (CPS) has been developed. The CPS is equipped with a diode laser at ~790 nm and two photo detectors, with a polarization plate in front of one of the detectors, to count the number of particles per second and to obtain the cloud phase information (i.e. liquid, ice, or mixed). The lower detection limit for particle size was evaluated in laboratory experiments as ~2 μm diameter for water droplets. For the current model the output voltage often saturates for water droplets with diameter equal to or greater than ~80 μm. The upper limit of the directly measured particle number concentration is ~2 cm$^{-3}$ ($2 \times 10^3$ L$^{-1}$), which is determined by the volume of the detection area of the instrument. In a cloud layer with a number concentration higher than this value, particle signal overlap and multiple scattering of light occur within the detection area, resulting in a counting loss, though a partial correction may be possible using the particle signal width data. The CPS is currently interfaced with either a Meisei RS-06G radiosonde or a Meisei RS-11G radiosonde that measures vertical profiles of temperature, relative humidity, height, pressure, and horizontal winds. Twenty-five test flights have been made between 2012 and 2015 at midlatitude and tropical sites. In this paper, results from four flights are discussed in detail. A simultaneous flight of two CPSs with different instrumental configurations confirmed the robustness of the technique. At a midlatitude site, a profile containing, from low to high altitude, water clouds, mixed phase clouds, and ice clouds was successfully obtained. In the tropics, vertically thick cloud layers in the middle to upper troposphere and vertically thin cirrus layers in the upper troposphere were successfully detected in two separate flights. The data quality is much better at night, dusk and dawn than during the daytime because strong sunlight affects the measurements of scattered light.



# 1 Introduction

Clouds play various roles in the weather and climate through latent heat transport, precipitation and the hydrological cycle, and shortwave and longwave radiative processes. Clouds are characterized by various microphysical parameters such as phase (liquid or ice), particle size distribution, number concentration, water content, and particle shape (e.g. Pruppacher and Klett, 1997), as well as various dynamical parameters (e.g. cloud cover, cloud type and vertical extent). There are various platforms and sensors for measuring cloud properties. Ground-based and satellite-borne remote sensing instruments mainly provide macroscopic features of cloud distribution, with some instruments giving estimates of microphysical properties, while in situ instruments on aircraft and meteorological balloons provide microphysical properties of clouds, often together with air temperature and humidity measurements from sensors on the same platform.

Meteorological balloons are low-cost devices for upper air sounding, and thus balloon-borne instruments are potentially useful for high-frequency measurements of cloud microphysical properties. Among such instruments, both the Hydrometeor Videosonde (HYVIS) (Murakami and Matsuo, 1990; Orikasa et al., 2013) and the balloon-borne Formvar replicators (Magono and Tazawa, 1966; Miloshevich and Heymsfield, 1997) continuously collect particles on a filmstrip during balloon ascent. HYVIS continuously transmits video images of the film surface to the ground, while the film with ice crystal replica is obtained upon recovery of the Formvar instrument. This type of particle collector requires knowledge of the collection efficiency on the film for quantitative interpretation of the results. In another instrument, the Videosonde (e.g. Takahashi, 1990, 2006; Suzuki et al., 2014), the primary component is a video camera with flash lamp that takes video images of falling particles introduced into the instrument; the studies by Takahashi also use an induction ring to measure the electric charge of the particles. The Videosonde has been primarily used for studies of precipitating clouds in the Asian monsoon region (Takahashi, 2006), while the HYVIS has been used for studies of midlatitude cirrus clouds (Orikasa et al., 2013; Orikasa and Murakami, 2015). The balloon-borne particle-measuring instrument, called the Backscattersonde (Rosen and Kjome, 1991), shines a flash-lamp beam periodically on the ambient air and measures the amount of backscattered light. This instrument has been extensively used for studies of polar stratospheric clouds (PSCs) and stratospheric aerosols (e.g. Suortti et al., 2001), but is also capable of measuring tropospheric clouds and aerosols. Based on the same principle, a much smaller instrument called the COBALD (Compact Optical Backscatter AerosoL Detector) was recently developed and used for cirrus studies (Brabec et al., 2012; Hasebe et al., 2013). The Backscattersonde and COBALD are used only at night because they measure artificial light scattered from outside the instrument. Finally, a balloon-borne optical particle counter (OPC) has been used for studies on PSCs (Hayashi et al., 1998) and on cirrus clouds and aerosols in the tropical tropopause layer (Iwasaki et al., 2007; Shibata et al., 2012). This instrument uses an inlet tube and a gear pump to introduce air into a scattering cell where a laser diode and a silicon photo diode are used to characterize the size distribution of aerosol particles.

The masses of the balloon-borne particle instruments described above range from 1 to 6 kg, except for COBALD whose mass is 500 g; this is much greater than the few hundred grams for recent operational radiosondes. Also, as these are special instruments developed by researchers, in most cases they are available only under collaboration with those researchers. In



practice, it is not easy to introduce and use these instruments as extensively as radiosondes and ozonesondes for both operational and research purposes.

In this paper, we present the results from a newly developed, small mass (~200 g) instrument called the Cloud Particle Sensor (CPS) flown in combination with a radiosonde. The CPS uses the air flow associated with the balloon ascent or descent to introduce air into a detection area within the instrument where a diode laser and two photo detectors are used to count the number of cloud particles and to characterize their phase. The CPS is based on a pollen sensor, the PS2, developed by the Shinyei Technology Co., Ltd. for a surface, multiple-point pollen-particle measurement network. We modified the PS2 for upper-air cloud-particle soundings, developed an interface board for the Meisei RS-06G radiosonde (Nash et al., 2011) and its upgraded version, Meisei RS-11G radiosonde, and made several laboratory experiments to characterize the capabilities and limitations of the instrument. We have made 25 test flights in total in Japan, Palau, and Indonesia during the period between 2012 and 2015. The remainder of this paper is organized as follows. The CPS and its interfacing with the Meisei radiosondes are described in Sect. 2. Results from four selected flights are described and discussed in Sect. 3, showing the capabilities and limitations of the CPS. Sect. 4 lists the main conclusions.

## 2 Instrumentation

### 2.1 Description of the CPS

Figure 1 shows a schematic diagram of the CPS. A near-infrared diode laser is used as a source of linearly polarized light. (The polarization is parallel to the scattering plane formed by the incident light, particle, and the two detectors.) The peak wavelength of the laser device has a nominal value of 790 nm, with individual devices varying between 775 and 810 nm, with a full width at half maximum of roughly 5 nm. Two silicon photo diodes are placed at angles of 55° (detector #1) and 125° (detector #2) with respect to the source-light direction. A polarization plate is placed in front of detector #2, so that detector #2 only receives light polarized perpendicularly to the source light. Lenses and slits are placed in front of the light source and the two detectors. The slit in front of the light source is 0.55 cm × 1.0 cm, while the slits in front of the two detectors are 0.50 cm × 1.0 cm. Because of the lens and slits, detectors #1 and #2 actually collect light scattered at 55° +/– 10° and 125° +/–10°, respectively. The cross section of the detection area that the detectors effectively see is estimated as ~1 cm × 1 cm, while its vertical extent is ~0.5 cm. Thus, the volume of the detection area is ~0.5 cm$^3$. The cross section of the air inlet is 1.0 cm × 1.2 cm. A particle that passes through the detection area scatters the laser light, and the two detectors receive the scattered light signal and thus count that particle.

The particle signal voltages from the two detectors ($I_{55}$ for #1 and $I_{125p}$ for #2) range from 0 V to 7.5 V with a resolution of 0.03 V. During the factory calibration, both are electronically adjusted to 2.5 +/–0.5 V using rough-surface particles of 30–40 μm diameter (with the polarization plate situated in front of detector #2). The rough-surface particles are used to calibrate the instrument after it is fully assembled. The original sampling frequency of the $I_{55}$ and $I_{125p}$ measurements is 11 kHz (i.e. virtually continuous), and a particle is detected when $I_{55}$ exceeds a threshold value of 0.3 V. Extensive laboratory





experiments were conducted to determine the lower detection limit of the particle size and the relationship between $I_{55}$ and water droplet diameter. Three CPS instruments were used to measure $I_{55}$ for 1 μm diameter polystyrene particles, 2, 5, 10, and 20 μm diameter borosilicate glass particles, and 30, 60, and 100 μm diameter soda lime glass particles. The CPS was unable to detect 1 μm diameter polystyrene particles but could detect 2 μm diameter borosilicate glass particles. Also, the

5 CPS often gives saturated outputs (~7.5 V) for 60 and 100 μm diameter soda lime glass particles. Mie scattering theory (Bohren and Huffman, 1998) gives the scattered light intensity as a function of the size and refractive index of a spherical particle, scattering angle, wavelength of the light, and other factors. This theory is used to convert the diameter of the particles used in the experiments to that of water droplets and spherical ice particles. In short, for the current model of CPS, the lower detection limit for water droplets is ~2 μm diameter, and the output voltage often saturates for water droplets equal

to or greater than ~80 μm diameter. See Appendix A for the detailed results of the experiments including the relationship obtained between $I_{55}$ and the diameter of water droplets and spherical ice particles.

To characterize the phase of the cloud particle (i.e. liquid or ice), the degree of polarization (DOP) is defined for this paper as

$$\text{DOP} = \frac{I_{55} - I_{125p}}{I_{55} + I_{125p}}.$$

DOP should be close to unity for spherical (i.e. liquid) particles because their $I_{125p}$ would be close to zero. On the other hand,

DOP would take various values between –1 and +1 for non-spherical ice crystals. The actual DOP distribution depends on the uncertainty in the factory calibration, the shape of measured particles, and the particle path in the detection area, among others. See Sect. 3 for the actual flight results. It should be noted that the characterization of particle phase using DOP might be possible even for large particles (i.e. >~80 μm) when $I_{55}$ is saturated. This is because for water droplets, $I_{125p}$ for those large particles would still be close to zero and thus DOP would be close to unity, while for ice crystals, DOP would take

various values. During the laboratory experiments discussed in Appendix A, it was confirmed that 60 and 100 μm diameter soda lime glass particles (corresponding to ~80 and ~130 μm diameter water droplets, respectively) give a DOP distribution with a peak located near unity, similar to that obtained for lower tropospheric water cloud measurements.

The CPS also monitors the particle signal width; i.e. the time interval of each particle signal (between the time when $I_{55}$ first exceeds 0.3 V and the time when $I_{55}$ falls below 0.3 V). The signal width data can be used for two purposes. One is to infer

the flow speed in the detection area whose vertical extent is ~0.5 cm. For the case of 5 m s$^{-1}$ flow speed, the expected signal width value is ~1 ms (milli-second), as the particle size (i.e. a few tens of μm) is usually negligible. The other is to monitor potential particle overlap in the case of dense cloud layers. Too long a signal-width value may indicate too many particles overlapping in the detection area and thus a substantial loss in particle count. In such a case, multiple light scattering would also occur to complicate the particle measurements ($I_{55}$ may be saturated, i.e. at ~7.5 V). See Sect. 2.3 for the full discussion

on the number concentration measurements by the CPS.

Finally, the latest version of CPS also monitors the direct current (DC) component for the detector #1 output, which represents any fixed light sources other than the particle scattering component. The DC component includes stray sunlight



and scattered (source or sun-) light from any dew and ice on the lenses and from the wall of the detection area. Therefore, high DC values during the daytime indicate potential contamination by stray sunlight.

The above description of the CPS is broadly applicable also to the pollen sensor PS2 from which the CPS was developed. The major exception is that for the PS2, the two detectors are placed at angles of 60° and –60° (the latter located below the light source) with respect to the source-light direction. The configuration of the PS2 gives symmetric geometry for the two detectors, but was found to cause particle contamination of the lower-side detector in cloud layers. Therefore, we modified the configuration to that shown in Figure 1. In Sect. 3.1 we will show the results from a simultaneous flight of a CPS with the PS2 configuration ("PS2 type") and the latest version CPS ("CPS1") with the configuration shown in Fig. 1. Other major modifications with respect to the PS2 included (1) the removal of a fan and the use of balloon ascent/descent to introduce the air into the detection area, (2) the installation of a small heater near the light source to reduce icing, and (3) the installation of slits in front of the laser diode and the two detectors to narrow the detection area and thus to increase the upper limit of the count number. Finally, the air inlet at the top of the CPS1 is 1.0 cm longer above the Styrofoam flight package surface to avoid counting particles detached from the package surface. For daytime soundings, additional longer air ducts were attached both at the top and the bottom to reduce contamination by stray sunlight. The dimension of the flight package is ~11.5 cm × 14 cm × 12 cm (Fig. 2), and the mass is ~200 g including the package and the three 3-V lithium batteries (CR123).

## 2.2 Interfacing with the Meisei radiosondes

The interface board developed for the CPS is capable of counting up to 1000 particles per second and providing $I_{55}$, $I_{125p}$, and the signal width values for up to 1000 particles. However, the downlink from the radiosonde to the ground-based receiver is limited to 25 byte $s^{-1}$ so the information needs to be compressed. The latest version, CPS1, provides the following data every second: (i) the number of particles counted per second; (ii) values of $I_{55}$, $I_{125p}$, and signal width for the first six particles coming into the instrument for each second; and (iii) the DC component for the detector #1 output. The CPS can be connected to a Meisei RS-06G radiosonde (Nash et al., 2011) or to its upgraded version, the Meisei RS-11G radiosonde. Figure 2 shows a photograph of the CPS1 within the flight package connected with the Meisei RS-11G radiosonde.

## 2.3 The number concentration measured by the CPS

The primary measurement of the CPS is the number of particles counted per second. If the flow speed within the detection area of the sensor is estimated or measured, the number per second can be converted to number concentration. The flow speed within an inlet duct is primarily determined by the balloon ascent rate (~5–6 m $s^{-1}$) but may also depend on the shape and the cross section of the duct and the ambient air density. At Moriya (36° N, 140° E), Japan at 17:11:47 LT (at dusk) on 23 November 2012, we launched a CPS with a hot-wire anemometer in an extended straight duct (i.e. 1.0 cm × 1.2 cm cross section) attached below the CPS. This anemometer had been calibrated by laboratory experiments under low-temperature and low-pressure conditions before the launch and its uncertainty is considered to be roughly 0.5 m $s^{-1}$. We found that in a cloud layer at 7–8 km, the balloon ascent rate was ~6 m $s^{-1}$, the flow speed measured by the anemometer was ~4–5 m $s^{-1}$,



and the flow speed estimated from selected, high-quality particle signal width data from the CPS was ~2–3 m s$^{-1}$. We also made another test flight at Moriya at 17:30 LT on 12 October 2011 (sect. 3.2.4 of Sugidachi, 2014) where the flow speed within four ~30 cm length straight ducts with different cross sections (i.e. three tubes with 2, 3, 15 cm diameter, and a duct with 2 cm × 2 cm inlet and 3 cm × 3 cm outlet) was measured using hot-wire anemometers during a balloon ascent. We

found that the flow speed in these ducts roughly agrees with the balloon ascent rate (~5–7 m s$^{-1}$) within ±1–3 m s$^{-1}$ in the troposphere. In summary, the assumption that the flow speed within the detection area is equal to the balloon ascent rate in the troposphere can be used as a first approximation but may cause uncertainty of a factor of ~2 in the estimated number concentration. (Note that the flow speed would be much smaller if longer, non-straight air ducts are attached both at the top and the bottom, e.g. for daytime soundings to avoid contamination by stray sunlight.) In this paper, for simplicity, we assume

a constant flow speed of 5 m s$^{-1}$.

As described in the previous section, the maximum number of particles (i.e. 1000) counted per second is determined by the interface board. However, the actual maximum number that can be correctly counted is thought to be much smaller because with such a high number concentration, particle signal overlap and multiple light scattering would occur. The following is a rough estimate of the actual limit for the number concentration and the number of particles counted per second. As described

above, the volume of the detection area is non-zero and estimated as ~0.5 cm$^3$. Therefore, when the particle number concentration is close to or more than 2 cm$^{-3}$, more than one particle would exist simultaneously in the detection area, resulting in particle overlap and multiple scattering and thus a counting loss. Therefore, the CPS can correctly measure number concentrations less than ~2 cm$^{-3}$ (i.e. $2 \times 10^3$ L$^{-1}$). Assuming that the flow speed is 5 m s$^{-1}$, and the cross section of the detection area is 1 cm$^2$, the number concentration of ~2 cm$^{-3}$ corresponds to ~1000 particles s$^{-1}$. Furthermore, the particle

signal width data can also be used to check whether particle signal overlap is occurring. As discussed in Sect. 2.1, the flow speed of 5 m s$^{-1}$ corresponds to a signal width of ~1 ms for a single particle. If the signal width value is close to or greater than ~1 ms, it is highly possible that signal overlap and counting loss are occurring. In actual flights, as will be shown in Sect. 3, dense cloud layers often give particle counts of ~100 s$^{-1}$ (or even less) and signal width of 1–5 ms; what happens here is that due to signal overlap, the number of particles is reduced to ~100 s$^{-1}$. When the signal width is greater than 1 ms, a partial

correction of the number of particles might be possible by using a factor $f$ = (signal width in ms)/(1 ms) and by considering the dimensions of the detection area (i.e. ~1 cm × 1 cm in the horizontal and 0.5 cm in the vertical) as $N_{cor} = N_{org} \times 2f \times 2f \times f$ = $N_{org} \times 4f^3$, where $N_{cor}$ and $N_{org}$ are the corrected and original numbers of particles, respectively. For example, when the signal width value is 5 ms, at least five particles would have been in line in the vertical in the detection area. The factor "$f$" is introduced for this reason. Furthermore, if it is assumed that particles are homogeneously distributed in the detection area,

the same argument would hold in the horizontal. The factor of 2 for the possible horizontal overlapping comes from the fact that the detection area is twice as long in the horizontal as the vertical. Although consideration of the statistical distribution of particles in the detection area may be needed for more accurate correction, this simple approach should give a first, rough approximation.



## 2.4 Relative humidity calculations

Before discussing the CPS results, some notes are given on the relative humidity (RH) plots in the following figures. The RH values reported from radiosondes are, by convention, always those with respect to liquid water even at air temperatures below 0°C. The ice saturation RH values below 0°C are always smaller than 100%, as shown by the dashed curve in the following RH figures. When the reported RH value is close to or even greater than the ice saturation RH value at the same location, it is highly possible that cloud particles were present. For the tropical soundings shown below (Sect. 3.3 and Sect. 3.4), we will also show the RH profiles obtained from the Cryogenic Frostpoint Hygrometer (CFH) (Vömel et al., 2007, 2016) flown on the same balloon (with the radiosonde temperature measurements). The CFH RH values in the tropical upper troposphere are considered to be much more reliable than the radiosonde RH values, because radiosonde RH sensors suffer seriously from time-lag errors in such high-altitude/cold-temperature regions (e.g. Nash et al., 2011). See Fujiwara et al. (2003) for the formula of the RH with respect to liquid water and of the ice saturation RH. In this study, we used the water vapour pressure equations developed by Hyland and Wexler (1983), as shown in Murphy and Koop (2005).

## 3 Results from test flights

Twenty-five test flights of the CPS with the Meisei RS-06G or RS-11G radiosonde were made during the period between November 2012 and March 2015. There were 11 flights at Moriya (36° N, 140° E), Japan; two at Tateno (36° N, 140° E), Japan; three at Palau (7° N, 134° E); and nine at Biak (1° S, 136° E), Indonesia. For the first two flights, at Moriya, we used a version of CPS with the same instrumental configuration as for the PS2 (the PS2 type CPS described in Sect. 2.1) and found that the lower detector part became highly contaminated after passing through supercooled cloud layers. We then developed the version (CPS1) with the configuration shown in Fig. 1 to minimize the particle contamination effects. We also tested some other versions with minor variations in the instrumental configuration to try to improve the upper limit of the number of particles. Among the 25 flights, two flights with two sensors on a single balloon were made at Moriya to compare different configurations (see Sect. 3.1 for the results from one of the flights). We also found that daytime soundings often give noisier CPS output data due to contamination by stray sunlight, and that soundings at dusk and at night give CPS data of much higher quality. In this section, we show the results from four balloon flights, two from Moriya (a midlatitude site) and two from Biak (a tropical site), and discuss the capabilities and limitations of the CPS.

## 3.1 Simultaneous flights of two different types of CPS

At Moriya, at 18:00 LT on 18 March 2013, we launched two CPSs simultaneously on a single 600-g balloon, one with the configuration shown in Fig. 1 (CPS1) and the other with the PS2 configuration (PS2 type; see Sect. 2.1), to compare the two types and to confirm the general performance of the CPS. The RS-06G radiosondes were used to downlink the data for both sensors and to measure the profiles of temperature, RH, height, pressure, and horizontal winds. At the time of launch, the site





was located at the southeastern edge of a low-pressure system, though there was no rainfall observed. Satellite cloud images show that the site was covered by a wide stratus cloud band associated with the low-pressure system.

Figure 3 shows the vertical profiles of radiosonde and CPS variables obtained from this flight. In the RH panel, the ice saturation RH curve (Sect. 2.4) is also shown. In the middle to upper troposphere, this curve, rather than 100%, indicates the reference to saturation and cloud formation. In the panel for number of particles, both original and corrected values (the latter is $N_{org} \times 4f^\beta$ when the particle signal width value (the maximum among up to six values per second) is greater than 1 ms, and $N_{org}$ when the width is equal to or smaller than 1 ms; see Sect. 2.3) are plotted. This panel also shows a conversion to approximate number-concentration values by assuming a constant flow speed of 5 m s$^{-1}$, with the cross section of detection area as 1 cm$^2$ (e.g. 1000 s$^{-1}$ corresponds to 2 cm$^{-3}$). The $I_{55}$, $I_{125p}$, DOP, and particle signal width data are those for the first six particles for each second, even when the number of particles per second is much greater than six, as described in Sect. 2.2. Also, DOP is not calculated when the corresponding $I_{55}$ is greater than 7 V; this is because in the ice or mixed-phase cloud regions, these cases create a strong peak around zero in the DOP frequency distribution. Finally, the DC component for the detector #1 output is also shown for this flight. Because this flight was made at dusk, there was virtually no contamination by stray sunlight. The average, background DC value (i.e. 1.06–1.1 for CPS1 and 1.2–1.24 for PS2 type for this flight) differs for individual CPSs due to, for example, different conditions of the wall of the detection area. As can be seen for this flight (at 5–6 km, around 8.8 km, and at 9.5–11 km), DC data can also show signals of this magnitude in dense cloud layers, where very strong particle overlap and multiple scattering occur so that the detector #1 output tends to be enhanced continuously (rather than intermittently). Thus, consideration of DC data might also be required for the full counting-loss correction. Under the influence of stray sunlight, DC data would show much greater values (e.g. more than 2 V).

In Fig. 3 the two CPSs show very similar cloud profiles. For example, the difference in the original number-of-particle values from the two CPSs is mostly within +/–10 s$^{-1}$ (with several exceptions for the cloud layers above 5 km). The characteristics of each cloud layer are as follows. From the surface up to ~2 km, there is a cloud layer of up to ~1 to 10 s$^{-1}$ (corresponding to ~$2 \times 10^{-3}$ to $2 \times 10^{-2}$ cm$^{-3}$ assuming a flow speed of 5 m s$^{-1}$), with high but unsaturated RH values. The $I_{55}$ values are generally less than 1 V, implying that the particles are very small (less than ~20 μm). The DOP values are mainly in the range 0.5–1 (see also Fig. 4), indicating that the particles are quasi-spherical and thus water droplets. This is also supported by the fact that the air temperature is higher than 0°C. The signal width values are generally less than 1 ms, indicating that particle overlap and counting loss are mostly not occurring. Between 5 km and 7 km, there is another cloud layer of up to ~$10^2$ to $10^4$ s$^{-1}$ (corrected values) (corresponding to ~$2 \times 10^{-1}$ to 20 cm$^{-3}$ assuming a flow speed of 5 m s$^{-1}$), with high RH values, some of which exceed the ice saturation RH values. The $I_{55}$ values range from ~0 to ~7.5 V, with many particles showing ~7.5 V, implying the existence of very large particles (>80 μm) and/or the occurrence of particle overlap. The DOP values are distributed widely within +/–1, indicating that the particles are non-spherical and thus ice particles. This is also supported by the air temperature ranging from –10°C to –30°C. The signal width values are mostly in the range 0–3 ms and sometimes reach ~5 ms (i.e. often exceed 1 ms), indicating that particle overlap and counting loss have occurred. Thus, the number-of-particle values have been corrected. From 8 km to the tropopause (~12.8 km), there is another group of



cloud layers with varying numbers of particles. There are three distinct peaks in (corrected) number of particles ($\sim 2 \times 10^3$ s$^{-1}$ at 8.7 km and around 10.5 km, and $\sim 10^3$ s$^{-1}$ at 12 km), with near saturated or even supersaturated RH values. The characteristics in $I_{55}$, DOP, and signal width are similar to those for the 5–7 km cloud layer. Thus, particle overlap and counting loss have also occurred for these upper tropospheric cloud layers, and the number-of-particle values have been corrected. These layers are also ice clouds, as indicated by the DOP distribution and air temperature values. In addition, the two CPSs also detected dust particles at least at the surface before the launch (not shown) and up to the first few hundred metres (e.g. the particles with the DOP of around zero). These dust particles are considered to be mostly of local origin due to the strong surface southerly winds associated with the low-pressure system whose centre was located to the northwest.

Figure 4 compares the DOP frequency distributions for two cloud regions, one at 0.5–1.5 km (i.e. water clouds since air temperatures are higher than 0°C) and the other at 12–13 km (i.e. ice clouds as air temperatures are lower than –65°C) and between the two CPSs with different configurations. Figure 4 shows that for water droplets, the DOP values only lie in the positive region, being mostly within 0.5–1, particularly for the CPS1. On the other hand, for ice cloud particles, the DOP is distributed widely within +/–1, with a peak around zero, particularly for the CPS1. It is noted that for water droplets, DOP does not necessarily become exactly unity as $I_{125p}$ does not necessarily become exactly zero. The latter is in part because there might be several azimuthal and zenith angles for the scattering water surface depending on the exact location of a water particle in the detection area. The actual DOP distribution for water droplets also depends on the factory calibration and thus differs slightly for different CPS instruments as shown in this and the following cases. However, as has been shown in this case, it is very different from that for ice particles.

In summary, the fact that the two CPSs with different configurations show very similar cloud profiles suggests the robustness of this technique for cloud particle measurements. Because some other earlier flights showed that the PS2-type configuration sometimes results in particle contamination to the lower-side detector, we have developed the CPS1 with the configuration shown in Fig. 1. The following three cases are all observed using the CPS1.

### 3.2 Midlatitude precipitating clouds

At Moriya, at 17:30 LT on 21 January 2015, the first commercial version of the CPS1 was launched with the RS-11G radiosonde to confirm its general performance under precipitating nimbostratus–stratocumulus conditions at midlatitudes. At the launch, low-level thick clouds were visually observed from the ground, and it started to rain lightly. A low-pressure system was developing to the southwest.

Figure 5 shows the vertical profiles of temperature, RH, and the CPS output data (same as in Fig. 3, but without DC data). The RH profile suggests that clouds covered basically throughout the troposphere, except for the first few hundred metres from the surface. It is suspected that the RH measurements may be ~10% RH biased wet at least from ~5 km to ~7.5 km (i.e. the pure ice-cloud region, as will be discussed below), but the vertical structure of the atmosphere is well captured. The CPS detected cloud particles basically throughout the troposphere from ~0.4 km up to ~7.5 km, and no particles above this level, generally consistent with the RH measurements. The profile of the corrected number of particles has less vertical structures



compared with the original one, indicating vales of $10^2$ to $10^4$ s$^{-1}$ (~2 × 10$^{-1}$ to 20 cm$^{-3}$). Above 1 km, the $I_{55}$ values of many of the particles are ~7.5 V, implying the existence of very large particles and/or particle overlap. It is these particles that give near zero DOP values (as both $I_{55}$ and $I_{125p}$ are ~7.5 V). Such particles are excluded from the analysis of DOP frequency distributions shown in Fig. 6. Interestingly, DOP data show a complex cloud-phase distribution in the vertical. In the region

up to ~1 km where DOP values are mostly 0.8–1.0, water clouds existed. From ~1 km up to ~4.5 km (about –20°C, capped by a distinct temperature inversion), DOP values show two groups, one with 0.8–1.0 and the other ranging widely within +/– 1.0. See Fig. 6b for the DOP frequency distribution at 2.5–4.5 km. This suggests the existence of mixed phase clouds where water and ice clouds co-existed. Above ~5 km, the DOP distribution is typical of ice clouds. Note that for this particular sounding, the DOP frequency distribution for water cloud particles is rather atypical, having its peak at 0.9–1 (Fig. 6a and

6b). Finally, the signal width values are 0–2 ms from the surface to 1 km and 0–5 ms above 1 km (i.e. often exceeding 1 ms), indicating that particle overlap and counting loss have occurred; thus, the correction has been made for the number-of-particle values.

In summary, this flight confirmed that the CPS can also be operated within midlatitude precipitating systems. This is also a good example showing the CPS's ability to detect mixed phase clouds as well as pure water clouds and pure ice clouds.

## 3.3 Tropical mid and upper tropospheric thick cloud layers

At Biak, Indonesia, at 18:00 LT on 23 February 2015, a CPS interfaced with an RS-11G and a CFH (Sect. 2.4) and ozonesonde interfaced with an RS-06G were flown on a single balloon during an observation campaign by the Soundings of Ozone and Water in the Equatorial Region (SOWER) project (e.g. Hasebe et al., 2013). Satellite cloud images show that the Indonesian Maritime Continent and tropical western Pacific were covered with groups of clouds associated with the

Madden–Julian Oscillation (e.g. Zhang, 2005), but there was no rainfall at the site at the launch time.

Figure 7 shows the vertical profiles of temperature, RH, and the CPS output data for the flight. Two RH profiles are shown, one taken from the RS-06G and the other calculated from CFH dew-point/frost-point temperature data and RS-06G air temperature data (see Sect. 2.4). From the surface to 5 km (~0°C air temperature), both RH values agree well. Between 5 km and ~12 km, detailed structures in the RH profile correspond well, though the RS-06G RH is ~5% RH wetter than the CFH

RH. Above ~12 km, the CFH RH shows layered structures in RH with a sharp drop at the cold point tropopause at ~15.8 km, while the RS-06G RH shows no such structures with only gradual decrease with height up to the tropopause. The latter is due to the time lag errors in the radiosonde RH sensors (e.g. Nash et al., 2011) in the upper troposphere and lower stratosphere, with very slow response at very cold temperatures. Thus, in the following discussion, we only refer to the CFH RH profile. The number of particles and the DOP in Fig. 7 show that ice cloud particles are observed continuously between

~5 km and the cold-point tropopause. The values of the corrected number of particles are ~$10^2$ to 5 × 10$^3$ s$^{-1}$, or 2 × 10$^{-1}$ to 10 cm$^{-3}$. The signal width values are mostly 0–3 ms, so that particle overlap and counting loss should have occurred; thus, the correction has been made for the number-of-particle values. Note that the CPS measurements were missing at 8.6–9 km for some reason.




In summary, this flight confirms that the commercial version of the CPS works well for ice clouds in the tropical middle and upper troposphere. It also shows that the system works well at cold temperatures down to –80°C (around the tropopause).

### 3.4 Tropical upper tropospheric thin cirrus layers

At Biak, at 18:00 LT on 27 February 2015, another set of a CPS and CFH–ozonesonde was launched during the same

SOWER campaign as in Sect 3.3. There was no rainfall at the site at the launch time. Satellite cloud images show that the site was still covered by groups of clouds similar to those on 23 February. The characteristics of temperature, CFH RH, and ozone above ~14 km (Fig. 8a, but ozone not shown) are indicative of downward stratospheric air transport in association with a downward-displacement phase of equatorial Kelvin waves (e.g. Fujiwara et al., 2001, 2009). Air was dry and cloud free above ~14 km for this reason.

Figures 8 and 9 show the results obtained from this flight. Between 0.8 km and 2.4 km, the air is saturated with a high number of water cloud particles of 1 to $10^4$ s$^{-1}$ or $2 \times 10^{-3}$ to 20 cm$^{-3}$ (corrected values). The signal width values are generally 0–2 ms with some larger values up to 4 ms. In the middle troposphere at 6–8 km, RH data from both the RS-06G and CFH show supersaturation, but the CPS shows zero particle count. There are two ice cloud layers in the upper troposphere, at 10.5–12.6 km and at 13.3–14 km, both with counts of 10 to $10^3$ s$^{-1}$ or $2 \times 10^{-2}$ to 2 cm$^{-3}$ (corrected values).

The signal width values are 0–5 ms for both layers, indicating that particle overlap and counting losses have occurred. For both layers, the air is saturated or even slightly supersaturated. It is interesting to note that a high supersaturation layer is located at 11.5–12.5 km (from the CFH measurements); i.e. in the upper part of the lower ice cloud layer. A possible interpretation is that the maximum cloud formation was occurring around 12–12.5 km where the degree of supersaturation is a maximum, but the sedimentation process created the peak cloud concentration at somewhat lower altitudes, around 11.3–

11.8 km. A similar observational result is shown in fig. 1 of Miloshevich et al. (2001).

During this SOWER campaign, the Mie depolarization lidar system at 1064 and 532 nm described in Shibata et al. (2012) was operated at Biak. On 27 February, the lidar data show that after 16:00 LT, a lower tropospheric cloud layer started to develop below 2 km, often preventing the lidar from measuring cloud layers in the middle–upper troposphere. Figure 10 shows the profiles of attenuated backscattering coefficient and volume depolarization ratio obtained from the lidar averaged

for 5 minutes over 18:55–19:00 LT when the lidar obtained relatively stable and good-quality data for the upper cirrus layers through the lower tropospheric cloud layer. The balloon was located at ~18 km at this time. The lidar data at 0–2 km were saturated and not suitable for scientific discussion. The lidar shows two distinct ice cloud layers at 10.5–12.5 km and at 13.5–14 km with non-zero depolarization of ~50% and ~60% (the latter with large variations), respectively (i.e. indicative of ice crystals). Also, the lidar does not show any clear cloud signals around 5–9 km. These results are qualitatively consistent

with the CPS measurements described above. Note that a close investigation of lidar data indicates that the thin water cloud layer at 5 km observed only by the lidar was quite variable in time (and thus inhomogeneous in horizontal). This is the reason why there was no cloud particle where and when the balloon passed 5 km.



In summary, the CPS in this flight successfully captured both lower tropospheric water clouds and upper tropospheric thin cirrus cloud layers in the tropics. Furthermore, the CPS measurements were consistent with the co-located lidar measurements in the middle–upper troposphere. Ground-based lidar systems are powerful tools for continuous cloud measurements (e.g. Fujiwara et al., 2009; Shibata et al., 2012). However, they have difficulties in measuring middle–upper

tropospheric clouds when there are dense cloud layers in the lower troposphere. Balloon instruments such as the CPS can be complementary or even essential for cloud measurements throughout the troposphere.

## 4 Conclusions

We have developed a balloon-borne small mass (~200 g) cloud particle sensor called the Cloud Particle Sensor (CPS), which is flown with a Meisei RS-06G or RS-11G radiosonde. The CPS is equipped with a near-infrared diode laser and two photo

detectors, one of which has a polarization plate in front of it. The CPS can detect particles larger than ~2 μm. The ambient air is introduced by balloon ascent/descent motion into the detection area within the instrument. The main output data from the CPS include the number of particles counted per second, and, for each particle, the scattered light signals from the two detectors ($I_{55}$ and $I_{125p}$ where $I_{125p}$ is the one from the detector with the polarization plate) and the particle signal width value. In practice, due to the limited radiosonde downlink speed, $I_{55}$, $I_{125p}$, and the signal width values are transmitted only for the

first six particles coming into the detection area each second. Using the $I_{55}$ and $I_{125p}$ values, we have defined the degree of polarization (DOP), which can be used to distinguish between water and ice cloud particles. The signal width values can be used to infer the flow speed in the detection area and to monitor potential particle overlap and counting loss. We have also made extensive laboratory measurements to determine the relationship between the $I_{55}$ value and water/ice diameter. For example, $I_{55}$ values of ~1.3 and ~2.2 V correspond to ~27 and 40-μm diameter water droplets, respectively (see Appendix A

for the details). It should be noted that even when $I_{55}$ is saturated (i.e. for large particles), the DOP distribution can still be used for cloud phase distinction.

If the flow speed in the detection area is known, the number of particles counted per second can be converted to the particle number concentration. Some test flights confirmed that the balloon ascent rate can be used as a first approximation, but this assumption may cause uncertainty of a factor of ~2 in the estimated number concentration. Furthermore, the maximum

number concentration that can be directly measured is considered to be limited by the non-zero volume of the detection area, which is ~0.5 cm$^3$. Thus, when the particle number concentration is close to or more than ~2 cm$^{-3}$, more than one particle would exist simultaneously in the detection area resulting in particle overlap and multiple scattering and thus a counting loss. The particle signal width value can be used to check whether particle signal overlap is occurring or not. If it is much greater than ~1 ms, particle overlap and counting loss are occurring. Even in such a case, a correction of the number of particles or

number concentration is possible by using the signal width value. In this paper, we used a simple approach for this correction. Future studies will pursue more accurate correction methods, including consideration of the statistical distribution of particles in the detection area.





We have conducted a total of 25 test flights between 2012 and 2015 at various midlatitude and tropical sites. In the paper we have shown the results from four flights that cover all the major aspects of the CPS performance. A dual flight of two CPSs with different instrumental configurations confirmed the robustness of the measurement technique and the improved performance of the CPS with the latest configuration. It was also confirmed that the CPS works very well under various

atmospheric conditions such as within midlatitude precipitating cloud systems and in the wide temperature range from +30°C down to –80°C in the tropics. The DOP values were found to be very useful for distinguishing between water clouds (DOP values distributed between +0.5 and +1) and ice clouds (DOP values widely distributed within +/–1). In one midlatitude flight there was a case where the DOP distribution was a combination of the water cloud and ice cloud distributions. Considering also the temperature range of this cloud layer (~0°C to ~ –20°C), it was concluded that the

observed cloud layer was a mixed phase cloud layer. Thus, the CPS is capable of detecting mixed phase clouds as well as water clouds and ice clouds.

The CPS is a small-mass instrument and suitable for flying with an operational radiosonde with a small weather balloon. It can detect cloud layers and their phase, including the mixed phase. It can also provide particle number concentration if the flow speed within the instrument is known. In dense cloud layers, however, a correction for counting loss is needed using the

signal width data (and probably the DC data as well). Developing a more accurate correction method is an area for future research. We have also provided the results from extensive laboratory experiments to determine the relationship between $I_{55}$ and the diameter of water or spherical ice particles. This information would be useful to further convert the number concentration information to water content information. Finally, it should be noted that because the CPS uses a light scattering method, daytime soundings often suffer from contamination by stray sunlight that gives noisier CPS output data. If

the CPS needs to be flown during the daytime, it is advised that some measures to reduce contamination by stray sunlight be considered, such as avoiding high solar elevation angles and installing longer (and non-straight) inlet and outlet ducts. (As the two detectors are looking down for the CPS1, additional duct installed at the bottom is also important.) It is also noted that if very long and/or non-straight ducts are used, the flow speed in the detection area might be much smaller than the balloon ascent/descent rate. The four flights described in this paper were all conducted at dusk.

**Appendix A**

Various types of standard spherical particles were used to determine the lower detection limit and the relationship between the $I_{55}$ value and the water/ice diameter for the latest version of the CPS (CPS1). Three CPS1 instruments (referred to as #1, #2, and #3 here) were used to evaluate instrument-by-instrument variability. Table A1 summarizes the information on the standard spherical particles used in the laboratory experiments.

Mie scattering theory (Bohren and Huffman, 1998) was used to convert the diameters of the standard spherical particles to those of spherical water droplets and those of hypothetical spherical ice particles. The refractive index values used for water were obtained from Hale and Querry (1997) by linear interpolation (e.g. $1.3300 + 1.4800 \times 10^{-7}$i at 775 nm and 1.3290 +



1.25 × 10⁻⁷i at 810 nm). For ice, we used the values obtained from Warren and Brandt (2008) (a table can be downloaded from http://www.atmos.washington.edu/ice_optical_constants/, accessed on 15 April 2016) by linear interpolation (e.g. $1.3054 + 9.39 \times 10^{-8}$i at 775 nm and $1.3047 + 1.4 \times 10^{-7}$i at 810 nm). We calculated the scattering matrix element $S_{11}$ and $S_{12}$ of Bohren and Huffman (1998) for scattering angles 55° +/–10° (integrated over this solid angle range) and for laser wavelengths between 775 and 810 nm at 1 nm step for each standard particle. The quantity ($S_{11} + S_{12}$) is proportional to the scattering intensity because of the parallel polarized incident light. We took the range of laser light wavelength to be 775–810 nm, though this is actually the instrument-by-instrument variability (see Sect. 2.1). For each standard particle, at each wavelength, we determined the diameter (or diameters) of water/ice particle that gives the same ($S_{11} + S_{12}$) value. Then, we took the minimum and maximum diameter values for the 775-810 nm range, which are shown in Table A2. The average of the minimum and maximum values are considered as the most probable value for simplicity.

Table A3 shows the results of the laboratory experiments. The number of particles measured ranges from ~230 to ~2700 depending on the instruments and the standard particles. It is found that the CPS1 is not sensitive to the 1 μm standard particles (i.e. ~1.36 μm water and ~1.42 μm ice particles), that the CPS1 gives ~0.6–0.8 V $I_{55}$ for 2–10 μm standard particles (i.e. ~2.10–13.52 μm water and 2.20–13.83 μm ice particles) without clear separation, and that the CPS1 gives ~1.33 and ~2.17 V $I_{55}$ for 20 and 30 μm standard particles, respectively (~26.65 μm water and ~28.30 μm ice, and ~39.50 μm water and ~42.29 μm ice, respectively). For 60 and 100 μm standard particles (i.e. ~79.78 and ~132.87 μm water and ~85.74 and ~143.49 μm ice), the CPS1 often gave saturated $I_{55}$ (53%–63% (83%–90%) of the 60 (100) μm particles gave more than 7.1 V $I_{55}$); as Table A3 shows, however, the average and standard-deviation values, calculated by including the saturated $I_{55}$ values as well, might still be useful to infer the water/ice particle diameters, at least statistically, for these ranges. Figure A1 shows the relationship between $I_{55}$ and the diameters of spherical water droplets and hypothetical spherical ice particles. This information can be used when $I_{55}$ values are converted to the diameters of spherical water/ice particles. In practice, a linear interpolation may be used for the $I_{55}$ range between 1.33 V and 5.36 V to obtain the diameter of water or ice particle. For the ranges below 1.33 V and above 5.36 V, the uncertainty would be high for the diameter estimation.

**Acknowledgements**

Balloon flights at Moriya, Japan, were made at the Meisei test field. Balloon flights at Biak, Indonesia, were made under the Soundings of Ozone and Water in the Equatorial Region (SOWER) project in collaboration with the Indonesian National Institute of Aeronautics and Space (LAPAN). Balloon flights were also made at Palau and at Tateno by the Hydrospheric-Atmospheric Research Center, Nagoya University and by the Japan Meteorological Agency, respectively, and their data contributed to the understanding of the CPS performance. This study was financially supported by the Japanese Ministry of Education, Culture, Sports, Science and Technology (MEXT) through Grants-in-Aid for Scientific Research (21244072, 22740306, and 26220101) and by the Institute of Space and Astronautical Science (ISAS) of Japan Space Exploration Agency (JAXA) through the Steering Committee for Space Science (SCSS).



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

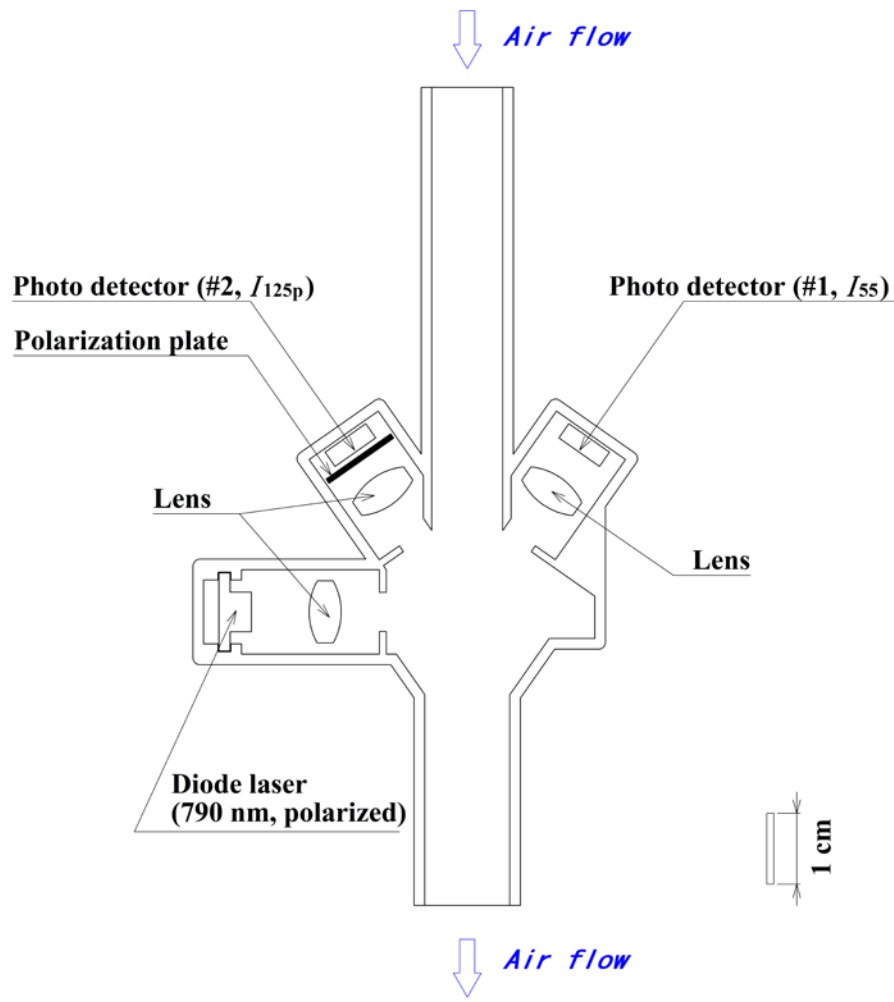

**Figure 1. Schematic diagram of the CPS (the latest version, "CPS1"). The diode laser, lenses, and two detectors (#1 and #2) are shown, together with the polarization plate for detector #2.**



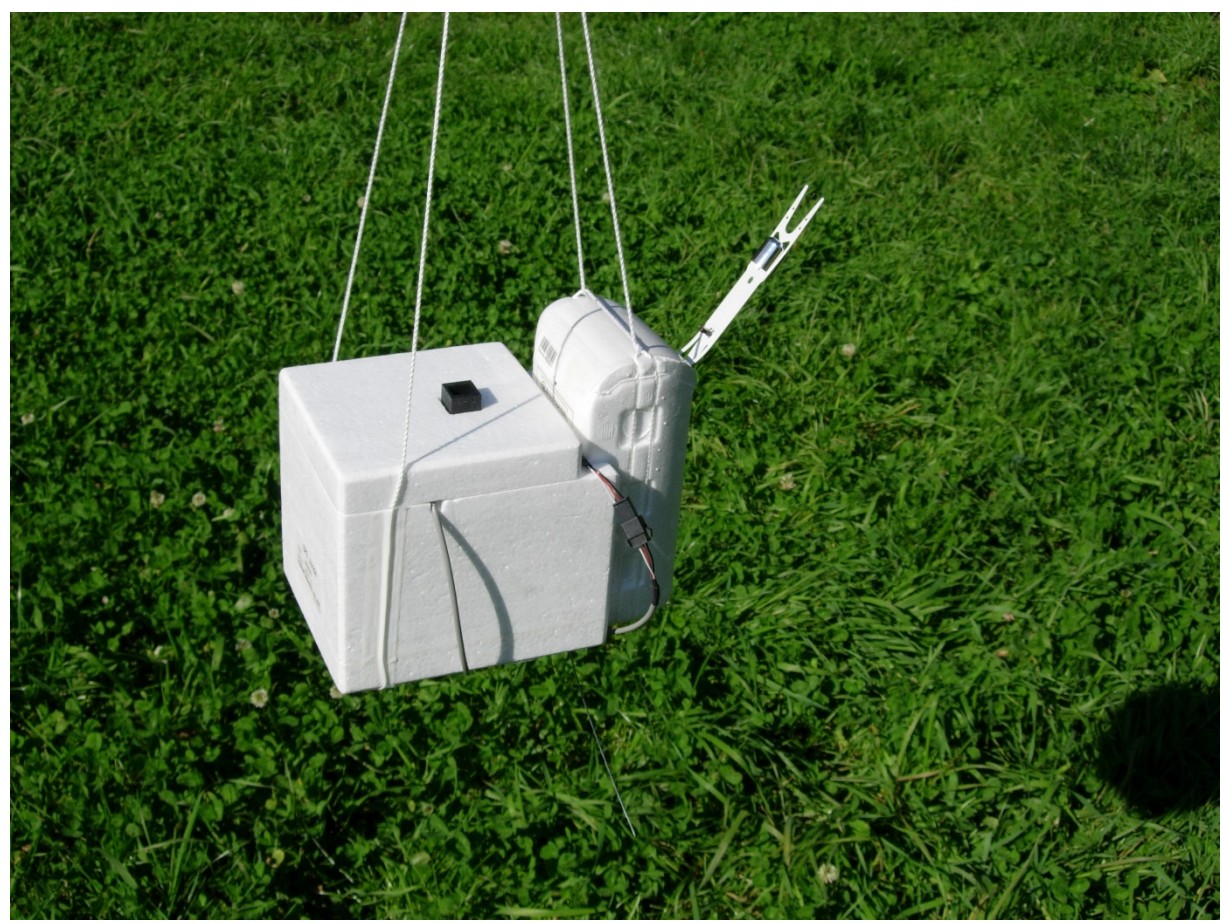

**Figure 2. Photograph of the CPS within its flight package (with a black inlet duct on top) connected to the Meisei RS-11G radiosonde (with the temperature and RH sensor boom). The three 3-V lithium batteries and the interface board are placed within the CPS flight package.**



**Figure 3. Results from a flight of two CPSs on the same 600 g balloon at Moriya, Japan, launched at 18:00 LT on 18 March 2013. Top panels (a–f) show data taken from the CPS1 (with the configuration shown in Fig. 1), and bottom panels (g–l) from the PS2-type CPS (see the last paragraph of Sect. 2.1). Vertical profiles are shown of (a, g) temperature (black), RH (blue solid), and ice**

5   **saturation RH (dashed blue) from the RS-06G radiosondes, (b, h) number of particles counted per second (and number concentration assuming 5 m s$^{-1}$ flow speed), original in black and corrected (see text for details) in red, (c, i) particle output voltages $I_{55}$ (red) and $I_{125p}$ (black), (d, j) the degree of polarization (DOP) (not calculated when $I_{55}$ is greater than 7 V; see text), (e, k) particle signal width in ms, and (f, l) the DC component from the detector #1 output.**





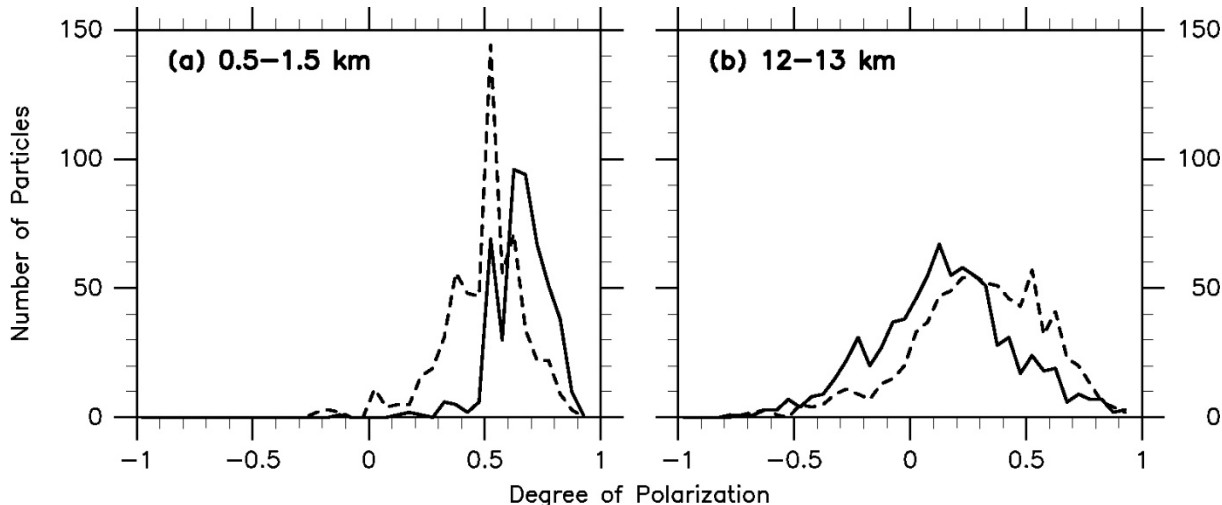

**Figure 4.** Frequency distributions of the degree of polarization (DOP) from the results shown in Fig. 3 at (a) 0.5–1.5 km (water clouds) and for (b) 12–13 km (ice clouds). For both panels, the solid curves are for the CPS1 (Fig. 1), while the dashed curves are for the PS2-type CPS (last paragraph of Sect. 2.1).





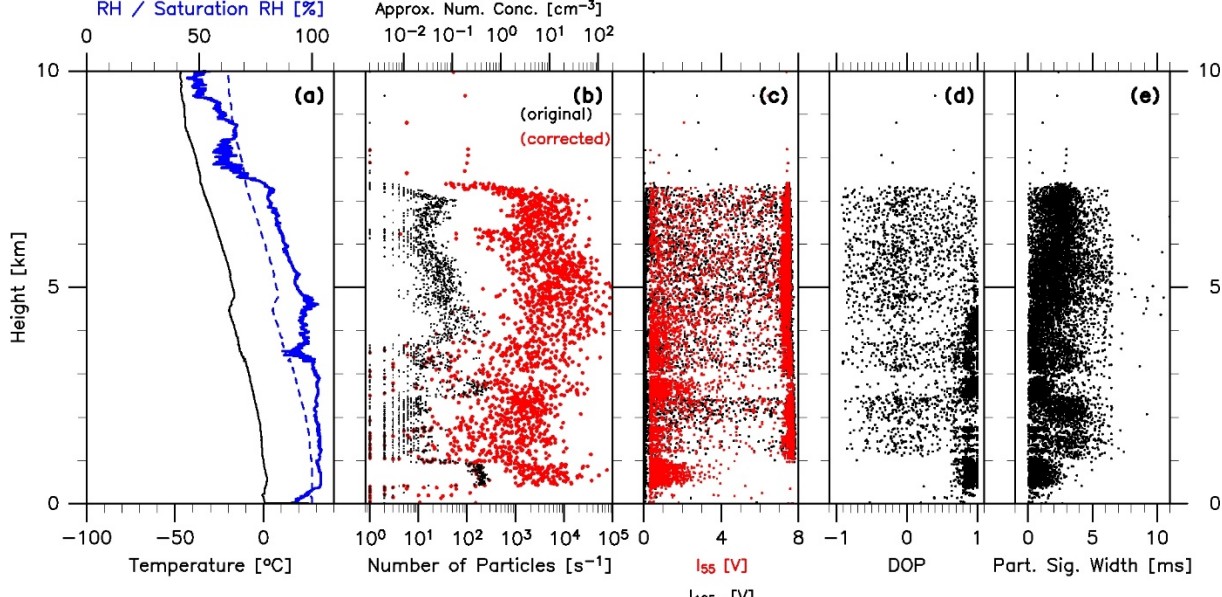

**Figure 5. Results from a CPS flight at Moriya, Japan, launched at 17:30 LT on 21 January 2015. Vertical profiles are shown of (a) temperature (black), RH (blue solid), and ice saturation RH (dashed blue) from the RS-11G radiosonde, (b) number of particles counted per second (and number concentration assuming 5 m s⁻¹ flow speed); original in black and corrected (see Sects. 2.3 and 3.1) in red, (c) particle output voltages $I_{55}$ (red) and $I_{125p}$ (black), (d) the degree of polarization (DOP) (not calculated when I55 is greater than 7 V; see Sect. 3.1), and (e) particle signal width in ms.**





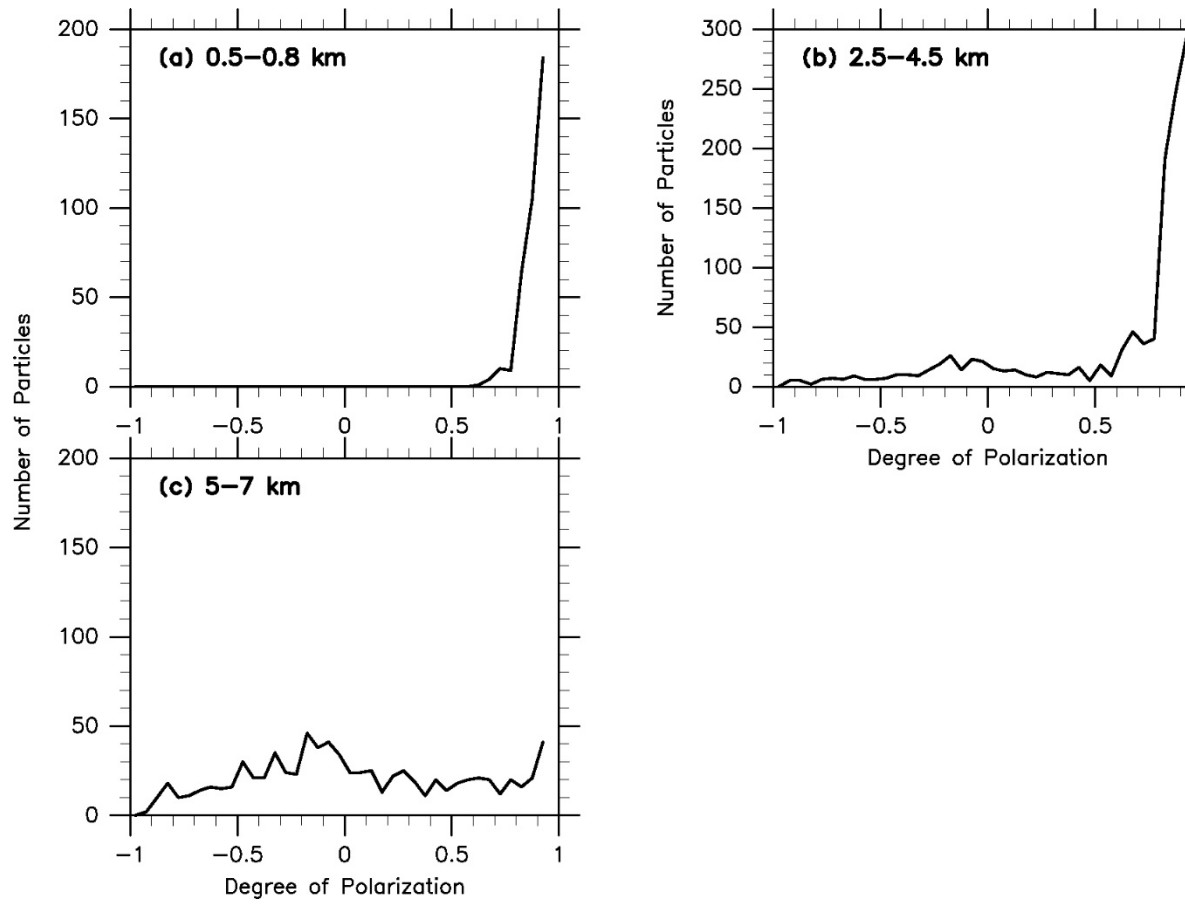

**Figure 6. Frequency distribution of the degree of polarization from the results shown in Fig. 5 at (a) 0.5–1.8 km (water clouds), (b) 2.5–4.5 km (mixed phase clouds), and (c) 5–7 km (ice clouds). Note that particles with $I_{55}$ values greater than 7.0 V were excluded from the analysis, as such particles create a very large peak around zero for ice or mixed-phase clouds.**





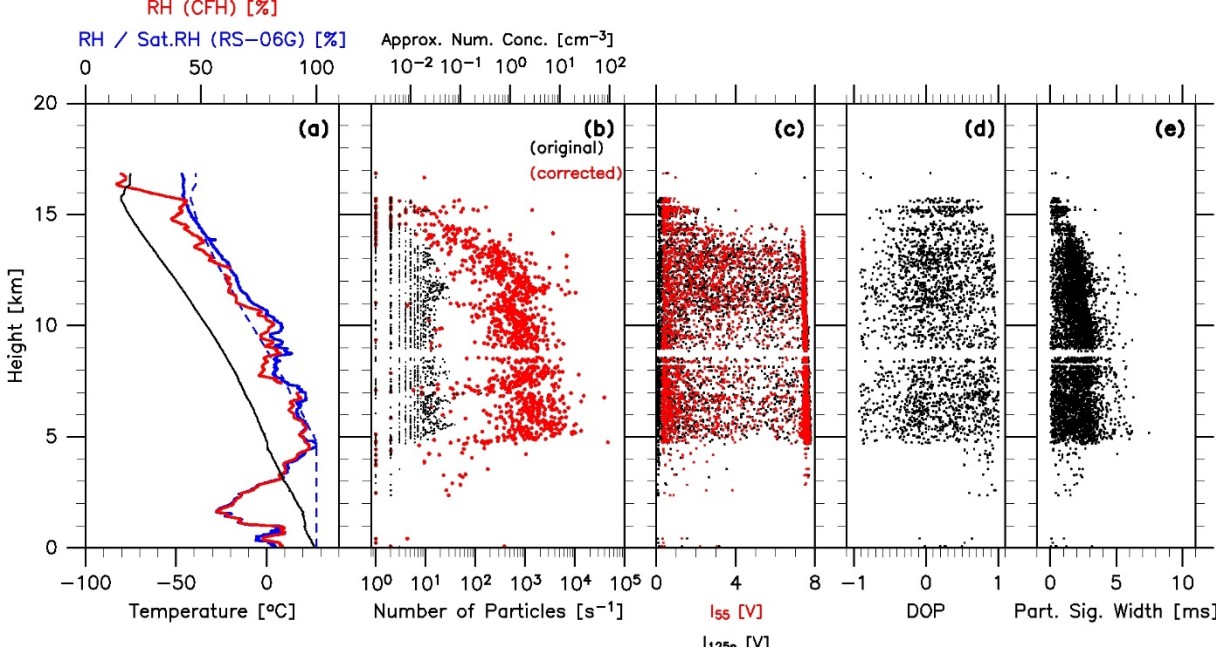

**Figure 7. As for Fig. 5, but for the profiles taken at Biak, Indonesia, launched at 18:00 LT on 23 February 2015. In panel (a), data from the RS-06G radiosonde are plotted, and RH from the CFH (red) is also shown.**





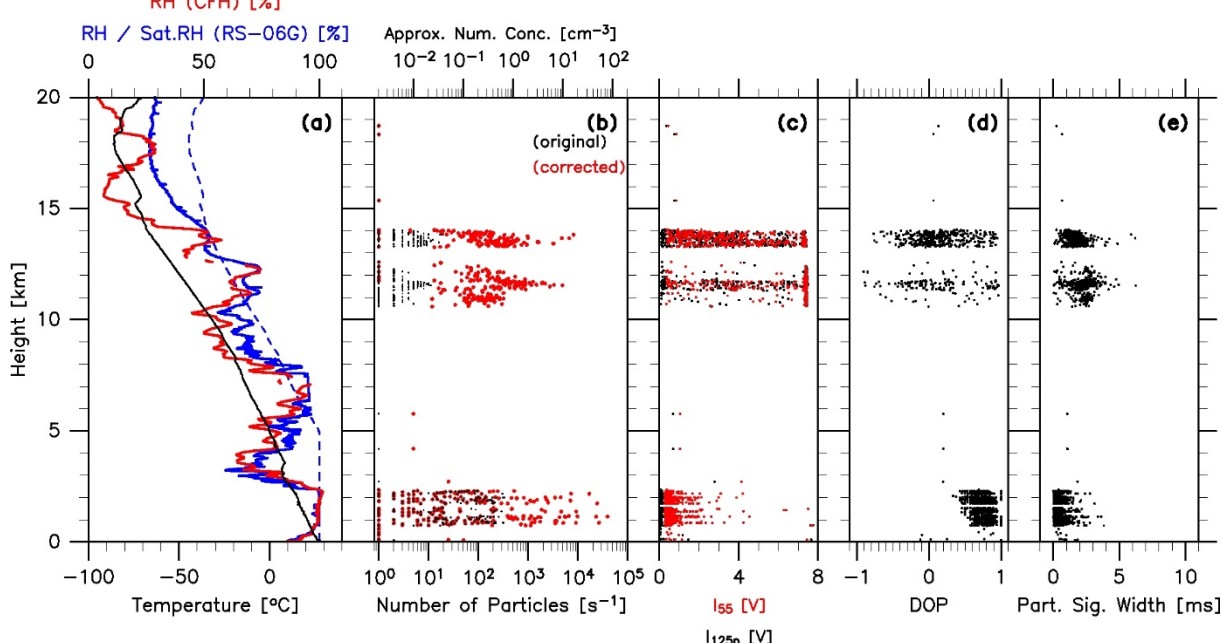

**Figure 8. As for Fig. 7, but for the profiles taken at Biak launched at 18:00 LT on 27 February 2015.**





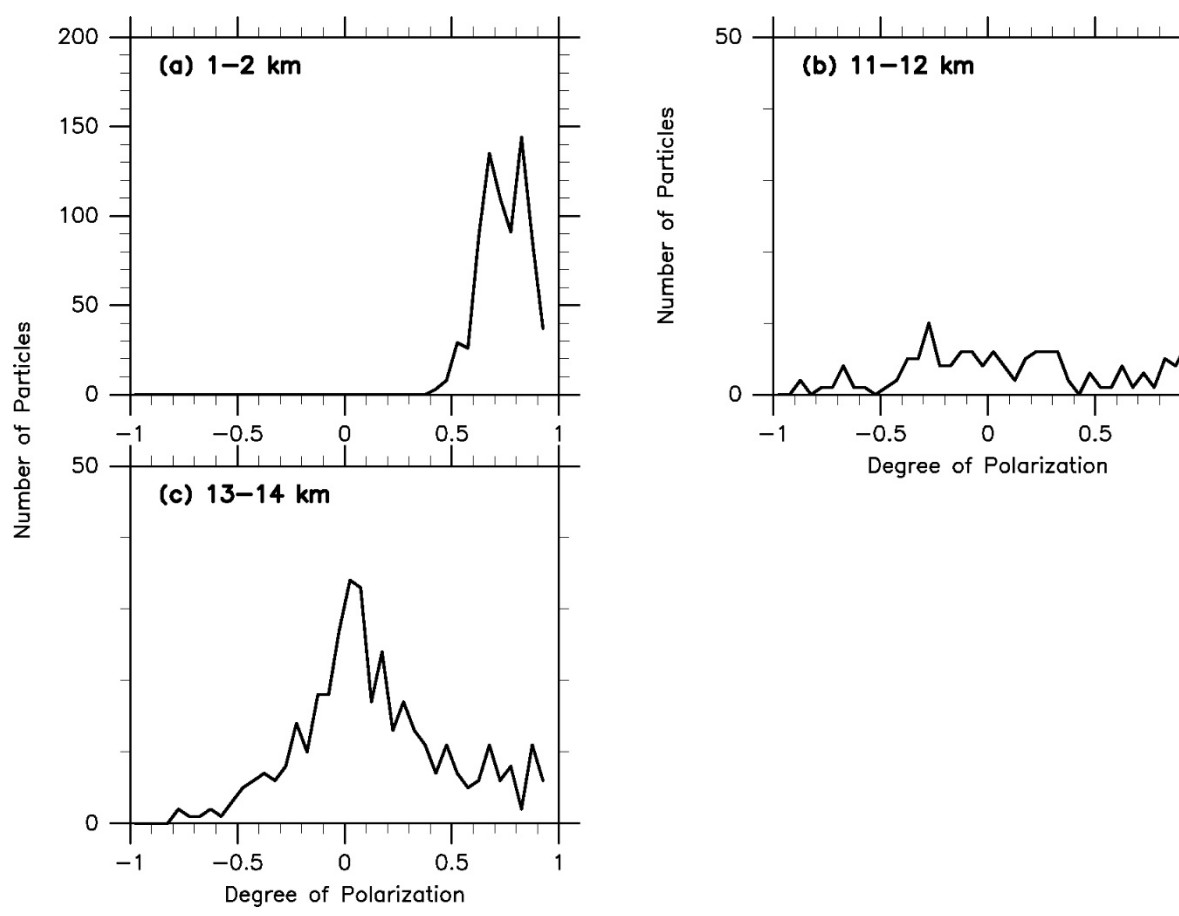

**Figure 9. As for Fig. 6, but from the results shown in Fig. 8 at (a) 1–2 km (water clouds), (b) 11–12 km (ice clouds), and (c) 13–14 km (ice clouds).**





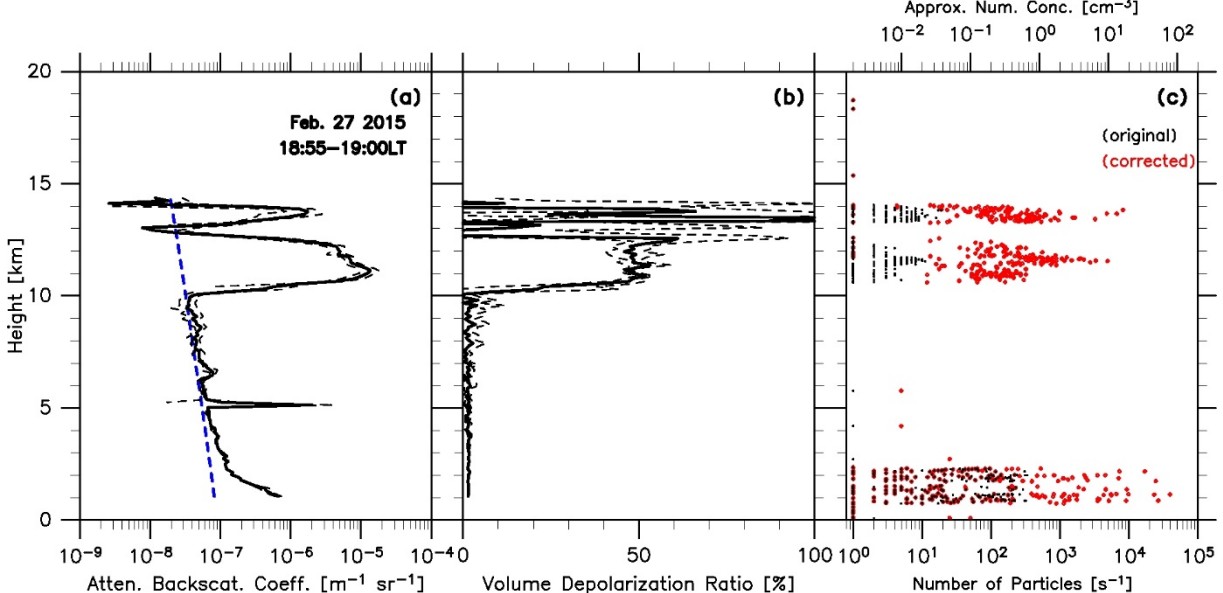

**Figure 10.** Vertical profiles of (a) backscattering coefficient (thick black) and (b) depolarization ratio (thick black) observed by a lidar system at Biak averaged over 18:55–19:00 LT on 27 February 2015, and (c) the same panel as Fig. 8b (i.e. quasi-simultaneous CPS measurements). The dashed blue curve in (a) indicates the estimated contribution of air molecules due to Rayleigh scattering. The thin dashed black curves in (a) and (b) indicate the standard deviation range. Lidar data at 0–2 km and above 14 km are not shown because signals were saturated for the former and data were highly noisy for the latter.



**Table A1. Standard spherical particles used in the laboratory experiments.**

| Diameter [μm] | Material | Refractive index[1] at 775 nm, and at 810 nm |
|---:|---|---|
| 1 | polystyrene | 1.5792 + 0.0i, 1.5778 + 0.0i |
| 2, 5, 10, 20 | borosilicate glass | 1.5359 + 0.0i, 1.5351 + 0.0i |
| 30, 60, 100 | soda lime glass | $1.5177 + 2.1495 \times 10^{-6}i$, $1.5170 + 2.6170 \times 10^{-6}i$ |

[1]**Refractive index information (as dispersion formula) was taken from http://refractiveindex.info/ (accessed on 25 February 2016).**
**For the borosilicate glass, the values for Schott ZERODUR glass were used.**



**Table A2. Conversion of the diameter of standard spherical particles to those of spherical water droplets and hypothetical spherical ice particles. In the third and fourth columns, the minimum, maximum, and the average of the two are shown in each table element.**

| Diameter of standard particles [μm] | Material | Diameters for water [μm] | Diameters for ice [μm] |
|---|---|---|---|
| 1 | polystyrene | 1.34, 1.38, 1.36 | 1.40, 1.44, 1.42 |
| 2 | borosilicate glass | 2.04, 2.16, 2.10 | 2.14, 2.26, 2.20 |
| 5 | borosilicate glass | 4.74, 7.12, 5.93 | 5.00, 7.42, 6.21 |
| 10 | borosilicate glass | 10.68, 15.82, 13.25 | 11.28, 16.38, 13.83 |
| 20 | borosilicate glass | 23.36, 29.94, 26.65 | 24.82, 31.78, 28.30 |
| 30 | soda lime glass | 36.78, 42.22, 39.50 | 39.50, 45.08, 42.29 |
| 60 | soda lime glass | 77.94, 81.62, 79.78 | 84.16, 87.32, 85.74 |
| 100 | soda lime glass | 130.14, 135.60, 132.87 | 141.74, 145.24, 143.49 |



**Table A3. Results from the laboratory experiments on the $I_{55}$ values for three CPS1 instruments (#1, #2, and #3) for various standard spherical particles. In each table element for individual CPS1 instruments and for the summary column, the average $I_{55}$ (V), its standard deviation, and the number of particles used in each experiment are shown. For the 10 and 20 µm standard particles, two sets of experiments were conducted on different days. The summary results in the fifth column have been obtained by combining the results from all the three instruments.**

| Standard particle [µm] | CPS1 #1 | CPS1 #2 | CPS1 #3 | Summary |
|---|---|---|---|---|
| 1 | (not sensitive) | (not sensitive) | (not sensitive) | (not sensitive) |
| 2 | 0.674, 0.0118, 869 | 0.678, 0.0224, 967 | 0.612, 0.0136, 1382 | 0.649, 0.0357, 3218 |
| 5 | 0.900, 0.0188, 1953 | 0.850, 0.0206, 2264 | 0.663, 0.0131, 2717 | 0.791, 0.106, 6934 |
| 10 | 0.650, 0.00853, 1382<br>0.809, 0.0128, 1399 | 0.673, 0.0156, 1246<br>0.752, 0.0195, 1205 | 0.691, 0.0157, 1210<br>0.721, 0.0169, 1595 | 0.717, 0.0554, 8037 |
| 20 | 1.45, 0.0455, 594<br>1.32, 0.0489, 462 | 1.34, 0.0535, 631<br>1.20, 0.0538, 476 | 1.38, 0.0523, 668<br>1.27, 0.0537, 595 | 1.33, 0.0929, 3426 |
| 30 | 2.39, 0.0634, 813 | 2.12, 0.0606, 887 | 1.93, 0.0739, 612 | 2.17, 0.194, 2312 |
| 60 | 5.59, 0.0767, 1053 | 5.41, 0.130, 432 | 4.57, 0.170, 331 | 5.36, 0.397, 1816 |
| 100 | 6.81, 0.0717, 502 | 6.36, 0.133, 271 | 6.68, 0.104, 282 | 6.66, 0.209, 1055 |





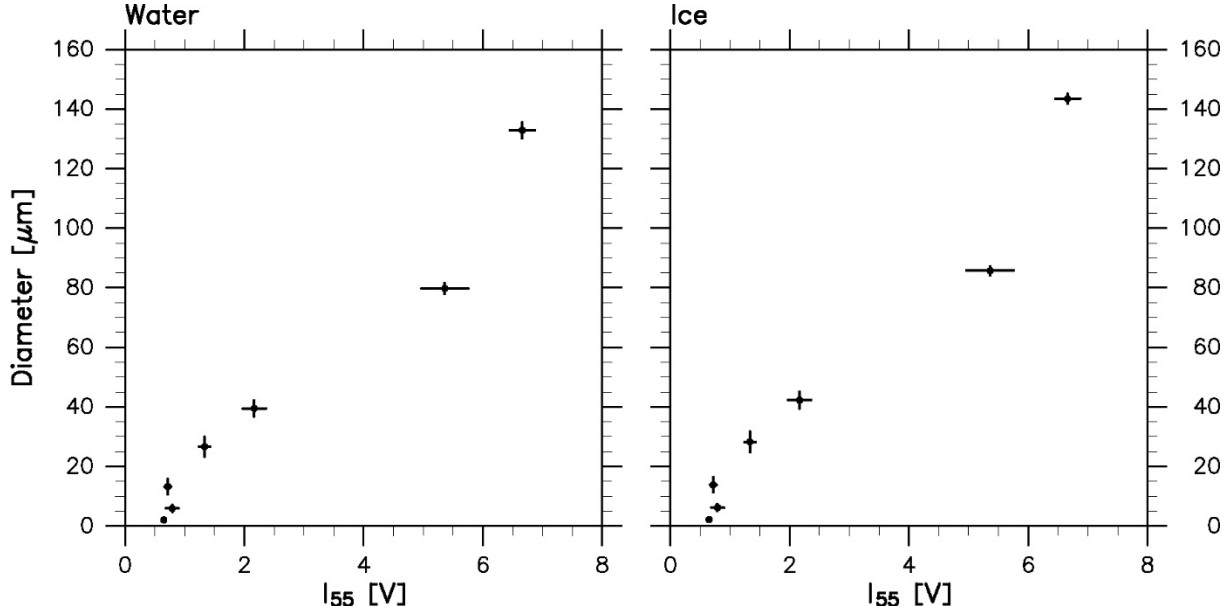

**Figure A1.** Relationship between $I_{55}$ and the diameters of spherical water droplets (left) and hypothetical spherical ice particles (right) based on Table A2 and the summary column of Table A3. The horizontal bars indicate the standard deviation range obtained from the laboratory experiments, while the vertical bars indicate the minimum and maximum diameters obtained from the Mie scattering calculations.