# Peer review of "Development of a cloud particle sensor for radiosonde sounding"

_Atmospheric Measurement Techniques, 2016_

## Referee Comment (RC1) · D. Baumgardner (Referee) · 22 Jun 2016

A single particle optical spectrometer, the CPS, is described for deployment on radiosondes that will count cloud particles and distinguish water droplets from ice crystals. The design is based on a commercial sensor that measures light scattered from aerosol particles and claims to separate pollen from other types of particles using polarization.

In my opinion, this paper is a long way from publication because it is missing some essential components that would make it a useful contribution: 1) scientific value, 2) Error analysis, 3) calibration details and 4) references to other instruments that measure the polarization state of cloud particles.

1) Scientific value The introduction discusses the importance of clouds and talks about

existing balloon borne cloud sensors but nowhere in the manuscript is there a discussion of either how balloon borne sensors contribute to the science or, in particular, how the sensor described in the paper, the CPS, will provide anything useful to our understanding of clouds. Based on a number of statements made throughout the paper, I don't think that the authors have a very strong knowledge of cloud physics. They refer to "dense clouds" as those with concentrations of 2 particles per centimeters cubed, the limit to their sensor's measurement capability. This is only the case for some cirrus clouds but certainly not for any low to mid-level water or mixed phased clouds.

Hence, the 2 cm3 coincidence threshold really makes this sensor useful only for cirrus clouds and you don't need a polarization detector to tell you that you are in all ice. This makes the CPS just a cloud detector with no useful information on the number concentration until it get high in the atmosphere.

In addition, the assumptions that are made about the difference between the polarization signal from water and ice are wrong. They state that the polarization by water droplets is nearly zero but if they do the calculations, they will discover that only at 180 degrees is this true. Water droplets will change the polarization state at angles less than 180 degrees and as a function of size. Hence, unless this bias is taken into account, the detection system as currently set up will classify some water drops as ice. This bias is clearly shown in Figs. 6a and 9a.

2) Error analysis No paper on measurement technique should be allowed publication without a serious error analysis and error propagation. The number "factor of 2" is given for concentration uncertainty but that is based on a very crude analysis. Given the many factors that affect the flow through the sample tube, the particle velocity can fluctuate much more than that. The suggestion that the transit time can be used to estimate the particle velocity is not valid. A number of things impact the transit time. If the authors would have shown a frequency histogram of transit times, I think it would show values varying by more than factors of three. The reason is that measuring the transit time from threshold to threshold makes the transit time sensitive to particles

size. Secondly, if the laser used has a Gaussian intensity profile, then the transit time will depend on where the particle passes through the beam and how far from the center of focus.

Nothing is said about how particles are constrained to go through the most intense portion of the laser beam. What is the sizing error due to passing through edges or away from the center of focus. Laser beams diverge so the scattering intensity from same sized particles can vary widely.

Is there a polarized filter in front of the laser to insure linear polarized light? If not, polarized laser diodes, particularly inexpensive ones, will have some fraction of the light in a different plane of polarization. This will bias the signal measured by the polarized detector

How is the sample volume defined? What is the uncertainty in the sample volume? Why use a sample volume that is at last 100 times larger than necessary and limits the concentration due to coincidence? Has this sample volume been mapped out?

3) Calibration "Rough particles" are mentioned as calibration for the polarization channel. What are rough particles?

Where are the calibration curves with water for I125? That would clearly show that there is a size dependent polarization signal for water droplets.

Where is the mapping of the sample volume?

4) Polarization references There are now a number of instruments that use measurements of the polarization ratio to differentiate spherical from non-spherical particle yet no mention is made of them. This is a significant oversight. There are a number of IN counters that do this, as well as the CAS-POL and CPSPD that differentiate water droplets from ice crystals looking at backscattered, polarized light.

In summary, unless the authors can better demonstrate that the CPS will provide any useful cloud property data, this paper should not be accepted for AMT.

---

## Referee Comment (RC2) · F.G. Wienhold (Referee) · 28 Jun 2016

This contribution presents a new lightweight balloon-borne Cloud Particle Sensor (CPS). It describes the instrumental setup with the constraints imposed by the conditions on operational soundings balloons, and explains how particle concentrations is estimated from the primary count rate. Results from test flights are presented that compare two instrumental setups and provide simultaneous measurements with the Cryogenic Frostpoint Hygrometer (CFH). Various midlatitude and tropical cloud layers are probed illustrating the deployment in a wide range of cloud types. This clearly demonstrates that different cloud types (pure water, mixed phase to cirrus) can be well identified and characterized.

The paper is well organized. The current development state of the new sensor and

the experience gained in various balloon sounding deployments deserve publication in Atmospheric Measurement Techniques. However, limitations and uncertainties need to be addressed more clearly, and a roadmap for future development should be added.

General comments

The term degree of polarization (DOP) is introduced unconventionally including differently adjusted detector sensitivities leading to an observed range from -1 and 1, whereas in common notion it would be expected to range from 0 to 1. To avoid confusion this needs to be stated more explicitly – see specific comments.

The air flow speed in the instrument is essential both to convert the count rate into number density, and to correct for signal overlap caused by multiple particle detection. Substantial flow speed reduction with respect to the ascent rate is reported for the instrument geometry used, rough agreement is found only for larger cross sections tested. Equating the flow speed with the typical sounding balloon ascent rate of 5 m/s – with a resulting reference signal width of 1 ms – is inappropriate when the effects number density conversion and overlap correction are not discussed (see also specific comments). Low number density regions near ground in Figures 5 and 7 with signal widths in excess of 1 ms conflict the undiminished flow assumption. The corrected particle count rate should be clearly addressed as an upper bound estimate, not just as corrected value.

The Meisei radiosonde interface is limited to 25 bytes/s while other radiosonde offer much larger bandwidth as 80 bytes/s for the Intermet Systems iMet or 100 bytes/s for Vaisala RS41. These would allow addressing the above problem by a better characterization of the signal width distribution instead of just transmitting the first six samples. I suggest adding a perspective for future development which could include real-time (on-board) data processing options in conjunction with increased downlink capacity. This could illustrate how the promising potential of this sensor could be exploited further.

Specific comments

Page 1, line 22: before "of the instrument", replace "volume of the detection area" by "ascent rate together with the detection volume and exposed cross section".

Page 3, lines 3-13: A very approximate cost might be indicated here.

Page 3, line 12: The situations covered by the four test flights could be mentioned here.

Page 3, lines 16 to 27: The orientation of the rectangular slits should be indicated explicitly in the text or the related Figure 1

Pages 3, line 30: do the "rough-surface particles" cause full depolarization and thus about equal sensitivity for both channels, or is the cross-polarized detector tuned more sensitive due to calibrating with only partially depolarized light?

Page 4, lines 6, 7: replace "gives" by "determines", and "other factors" by "polarization".

Page 4, line 15 (compare comment p 3, l 30): To not confuse the reader it should be stated that for equally sensitive detector the DOP should be between 0 and 1. As detector #1 senses the sum of co- and cross-polarized scattered light, negative DOP values can only result from higher sensitivity of detector #2.

Page 4, line 16: mentioning "the uncertainty in the factory calibration": Is there clear calibration target?

Page 4 line 20: Replace "various" by "nonzero".

Page 4, lines 26, 27: It should be stated that particle overlap excludes flow rate determination: the two signal-width interpretations are complementary.

Page 5, line 12: Replace "is 1.0 cm longer" by "extends 1.0 cm".

Page 5, line 19: Other radiosondes offering external instrument interfaces provide higher bandwidth, e. g. 80 bytes/s (Intermet iMet-1RS) or 100 bytes/s (Vaisala RS41). This additional capacity could be used in the future development perspective.

Page 6, line 10: The experimental findings described in this section confirm that the

flow rate through the instrument is reduced, as expected. With the tested larger cross section the flow agrees with the ascent rate, but for the configuration used in what follows it is only about half the balloon ascent rate. It should be stated that assuming the nominal flow speed of 5 m/s implies under-estimating both the count rate and the 1 ms reference signal width.

Page 6, line 27: The signal width factor f enters the count rate correction to the third power. An assessment of the consequences should be provided.

Page 7, line 5: Replace "is close or even greater" by "exceeds", and delete "highly" in line 6: Ice saturation between 120% and 130% are frequently observed without cloud formation.

Page 8, line 10: Description of future work could be anchored here which allows for an improved signal width characterization rather than just sending the first six samples.

Page 8 line 11: Not calculating the DOP for signals exceeding 7 V contradicts the statement on page 4, lines 17 to 22.

Page 9, line 24: What is the major difference of "the first commercial version" flown 2015 versus the instrument launched 2013? Near ground level signal width of Figure 5 appears to have increased with respect to Figure 3.

Page 10, line 3: Excluding DOP analysis again contradicts page 4, lines 17 to 22.

Page 10, lines 10-11 and 31: The authors set a (debatable) signal width limit of 1 ms as onset for particle overlap correction (see p 6, l 24). This should be reflected using 1 instead of 0 as lower limit in the ranges given for the signal width.

Page 10 line 33: "for some reason" needs further specification: instrumental or telemetry failure?

Page 11, line 15: See signal width range comment for page 10.

Page 11, lines 16-20: Supersaturated layers above cirrus cloud tops are not surprising

and have been discussed in detail in Brabec et al. (2012) , including microphysical modeling.

Page 11, line 21: Which of the lidar wavelengths mentioned is used for analysis and displayed in Figure 10?

---

## Referee Comment (RC3) · Anonymous Referee #3 · 3 Jul 2016

This paper describes the new balloonborne instrument of cloud particles whose signal data is transmitted with a radiosonde to a ground station. The new sensor, CPS has capabilities to count particles larger than about 2 microns and to distinguish between cloud water droplets and ice crystals. The authors also mention some limitations using the test flight data in several launch sites.

However, it is not clear what levels of accuracy and range of uncertainties the new sensor has during the ascent flight. Although it might be the first light-weight cost-effective balloonborne sensor over the cloud particle size range with a polarization function, the more quantitative evaluations are required for detecting/measuring variety of atmospheric particles.

I am ready to recommend the publication after the authors revise the manuscript taking

into account the following comments, especially in terms of more supporting facts and figures which clearly indicate the measurement uncertainties and limitations. In my opinion, the manuscript will require a major revision for publication.

General comments:

Even if the main purpose of the paper is the demonstration of the new sensor, not quantitative discussions in detail, the basic performance of the sensor under the various atmospheric conditions is necessary since the CPS is a balloonborne sensor, rather than a ground-based instrument.

The manuscript is well written, however, I did not fully understand the strong points of this paper, compared to the current airborne instruments of cloud particles. There are many other airborne instruments for cloud and aerosol particle measurements with the polarization function: for instance, CASPOL (Baumgardner et al., 2001, 2011; Glen and Brooks, 2013; Nichman et al., 2016), SID (Hirst et al., 2001; Cotton et al., 2010), and CPSPD (Baumgardner et al., 2014). These instruments have a higher sensitivity with more detectors, and those should be referred in the paper at least. Also, a polarization optical particle counter has developed and demonstrated (Kobayashi et al., 2014), although it covers the different measurement size range (0.5 – 10 microns) and is the ground-based instrument. Therefore the comparison in laboratory experiments with other sensor or instrument is strongly recommended to exhibit the basic performance of polarization measurements.

If the CPS has an advantage of cloud phase determination in natural clouds, it remains ambiguous only from the frequency distribution of the degree of polarization (DOP). Thinking of the ice initiation from supercooled water clouds, not so many particles are detected as ice since the number of natural ice nuclei is generally scarce in the troposphere. So it is quite difficult to detect the signal of few ice crystals from the DOP frequency distribution. Even in the water clouds, there is certain amount of ranges of DOP, seeing from the Figs. 6a and 9a (I think that Fig. 4a must indicate the signals of

aerosols, not water droplets. Please see the specific comment below.).

Or if the CPS has an advantage of counting of particles, the number concentrations in clouds, the sampling volume and counting efficiency are more thoroughly evaluated. To evaluate the sampling volume and its uncertainty during the ascent flight is essential to estimate the number concentration of particles. In the paper, only the rough estimate of detection area, ∼0.5 cm3, is written in the text. Only the fixed typical ascent rate, ∼5 m/s, is used, although it will be variable from the surface to upper levels even in the same case.

Is the real counting efficiency considered to be uniform inside the cross section of detection area (1cm x 1cm)? Even if the intensity profile of a laser device and the air flow inside the area are assumed to be uniform, in fact, the efficiency and uncertainty of counting particle might not be distributed with uniformity between in the center and near the edge of the detection area. Does the intensity and peak wavelength of the laser change during the ascent or under different ambient conditions? If so, does it influence the measurement accuracy at what amount of uncertainties?

Moreover, how are the capabilities of measuring coarse aerosols (for instance, dust or sea-salt particles) in real atmosphere? In other words, does the CPS have the capability to distinguish among water droplets, ice crystals, and various types of coarse aerosols over the micron-sized range?

Specific comments:

The following comments do not cover all general points above, but explain specifically as well as some minor points.

Page 3, Lines 19-20: Why the two photo detectors are placed at angles of 55 and 125 degrees? Do these set of angles have the best performance to distinguish between water droplets and ice crystals?

Page 3, Lines 29-31: As for the calibration or ground test before the launch, in what

ways is the sensitivity of both channels confirmed to be equal in terms of size measurements? In what ways is the calibration of polarization signal done quantitatively using the rough-surface particles? And the information on the rough-surface particles is required in the text (commercially available? mean size and its standard deviation? refractive index?).

Page 4, Lines 1-11: The paper describes the laboratory data focused on the relationship between I55 and the particle diameter. The performance of I125 is also needed to be examined to show the accuracy and uncertainty.

Page 6, Line 5: Using the hot-wire anemometer sensor data, the flow speed in the duct is compared with the balloon ascent rate. Thinking of the outlet area (3cm x 3cm) is wider than the inlet area (2cm x 2cm), what is the mechanism of significantly higher speed in the duct in some cases?

Page 6, Lines 11-18: The actual maximum number counted per second is thought be much smaller than the estimated maximum number ($\sim$1000) from limitation of the interface board. What was the maximum number per second measured from the test flight data (in water clouds)? The upper limit of number concentrations in the CPS measurements can be much smaller than $\sim$2 cm-3? The typical number concentration of cloud droplets in natural clouds covers from tens to thousands of particles per cubic centimeter, so basically is it difficult to reliably measure the number concentrations of water clouds even with development of any correction algorithms?

Page 8, Lines 22-23: The manuscript states that there is a cloud layer from the surface to $\sim$2km. However it is considered more reasonable that the signal should indicate the layer of coarse aerosols (possibly in swollen state) since the RH is subsaturated with respect to water and it is inconceivable to have very low concentrations for cloud water droplets in the subsaturated layer. That is why I think it is quite difficult to determine the cloud phase only from the DOP frequency distribution.

Page 11, Lines 21-29: Although it might be impossible to compare the figures related to

depolarization between the CPS and the remote-sensing measurements, is there any way to evaluate the DOP uncertainty? The comparison with other airborne sensors in laboratory experiments is useful to show the CPS performance for depolarization.

Pages 13-14 (Appendix A): The laboratory experiments are described using the standard spherical particles with different types and sizes. The description on how to enter the particles into the detection area is also required in the text. Is the air vacuumed from the bottom of the particle inlet duct at 5 m/s? How do the particles dispense before entering the particle inlet? Are there any calibration data to be described using non-spherical particles? That might be helpful to show the uncertainty of the CPS polarization measurement.
* * *

---

## Author Comment (AC1) · 2 Sep 2016

**Reply to Referee #1 (Dr. D. Baumgardner)**

Thank you very much for your reviewing our manuscript and providing us valuable comments and suggestions.

*A single particle optical spectrometer, the CPS, is described for deployment on radiosondes that will count cloud particles and distinguish water droplets from ice crystals. The design is based on a commercial sensor that measures light scattered from aerosol particles and claims to separate pollen from other types of particles using polarization.*

*In my opinion, this paper is a long way from publication because it is missing some essential components that would make it a useful contribution: 1) scientific value, 2) Error analysis, 3) calibration details and 4) references to other instruments that measure the polarization state of cloud particles.*

*1) Scientific value*
*The introduction discusses the importance of clouds and talks about existing balloon borne cloud sensors but nowhere in the manuscript is there a discussion of either how balloon borne sensors contribute to the science or, in particular, how the sensor described in the paper, the CPS, will provide anything useful to our understanding of clouds.*

The weather balloon with a radiosonde is one of the major platforms for upper air sounding, measuring vertical profiles continuously from the surface up to the middle stratosphere in ~100 minutes. The radiosonde temperature and humidity measurements, including those from balloon-borne special hygrometers, have been well characterized by regular and special intercomparison campaigns (e.g. Nash et al., 2011; Vömel et al., 2016). The typical balloon ascent rate is ~5 m s$^{-1}$, which is much smaller than the speed of aircrafts, easing some of the technical challenges that aircraft instruments may face (e.g. shattering of ice crystals; Baumgardner et al., 2012). On the other hand, balloon instruments need to be disposable, operable with small batteries, and of small mass. These factors may result in technical challenges when developing balloon-borne cloud particle instruments. As described in Introduction, the masses of existing balloon-borne cloud particle instruments range from 1 to 6 kg except for COBALD whose mass is ~500 g. The CPS is a balloon-borne instrument with a small mass (~200 g) which is comparable to that of modern radiosondes, and thus is much more easily combined with a radiosonde to detect the "existence" of cloud particles. This may provide advances in understanding of cloud and water vapour processes because this gives us more opportunities to obtain a set of vertical profiles of temperature, humidity, and (existence of) cloud particles compared to the

existing much heavier balloon-borne instruments. A combination of radiosonde temperature and humidity measurements with a ground-based or satellite particle remote-sensing measurements (as discussed in Section 3.4, for example) may be another way to obtain such a data set, but the measurement collocation is always an issue particularly for high-altitude cirrus clouds (e.g. Shibata et al., 2007). As a disposable instrument, there is limitation in number concentration measurements, though we propose a simple, partial correction algorithm and a simple algorithm to convert the number of count to number concentration. Furthermore, utilizing the polarization information, the CPS can distinguish between water, ice, and mixed phase cloud layers (i.e. for a group of particles; even for a single particle for particular conditions); more detailed explanation on this is given below.

Some of the authors of this paper are considering the following specific applications: subvisual cirrus cloud processes in the tropical tropopause layer (TTL) (e.g. Iwasaki et al., 2004, 2007; Fujiwara et al., 2009; Shibata et al., 2007, 2012; Inai et al., 2012); radiosonde relative humidity sensor validation during radiosonde intercomparison campaigns; applications for dropsonde systems (by modifying the CPS1) to observe precipitating clouds (e.g. in association with typhoons); applications for long-duration balloon systems flying in the UTLS; among others. We believe that other researchers would have more ideas to utilize this instrument.

References:

Baumgardner, D., Avallone, L., Bansemer, A., Borrmann, S., Brown, P., Bundke, U., Chuang, P. Y., Cziczo, D., Field, P., Gallagher, M., Gayet, J.-F., Heymsfield, A., Korolev, A., Krämer, M., McFarquhar, G., Mertes, S., Möhler, O., Lance, S., Lawson, P., Petters, M. D., Pratt, K., Roberts, G., Rogers, D., Stetzer, O., Stith, J., Strapp, W., Twohy, C., Wendisch, M.: In situ, airborne instrumentation: Addressing and solving measurement problems in ice clouds, Bull. Amer. Meteorol. Soc., 93 (2), ES29–ES34, doi:10.1175/BAMS-D-11-00123.1, 2012.

Fujiwara, M., Iwasaki, S., Shimizu, A., Inai, Y., Shiotani, M., Hasebe, F., Matsui, I., Sugimoto, N., Okamoto, H., Nishi, N., Hamada, A., Sakazaki, T., and Yoneyama, K.: Cirrus observations in the tropical tropopause layer over the western Pacific, J. Geophys. Res., 114, D09304, doi: 10.1029/2008JD011040, 2009.

Inai, Y., Shibata, T., Fujiwara, M., Hasebe, F., and Vömel, H.: High supersaturation inside cirrus in well-developed tropical tropopause layer over Indonesia, Geophysical Research Letters, 39, L20811, doi: 10.1029/2012GL053638, 2012.

Iwasaki, S., Tsushima, Y., Shirooka, R., Katsumata, M., Yoneyama, K., Matsui, I., Shimizu, A., Sugimoto, N., Kamei, A., Kuroiwa, H., Kumagai, H., and Okamoto, H.: Subvisual cirrus cloud observations using a 1064-nm lidar, a 95 GHz cloud radar, and radiosondes in the warm pool region, Geophys. Res. Lett., 31, L09103, doi:10.1029/2003GL019377, 2004.

Iwasaki, S., Maruyama, K., Hayashi, M., Ogino, S.-Y., Ishimoto, H., Tachibana, Y., Shimizu, A., Matsui, I., Sugimoto, N., Yamashita, K., Saga, K., Iwamoto, K., Kamiakito, Y., Chabangborn, A., Thana, B., Hashizume, M., Koike, T., and Oki, T.: Characteristics of aerosol and cloud particle size distributions in the tropical tropopause layer measured with optical particle counter and lidar, Atmos. Chem. Phys., 7, 3507–3518, doi:10.5194/acp-7-3507-2007, 2007.

Nash, J., Oakley, T., Vömel, H., and Li, W.: WMO Intercomparison of high quality radiosonde observing systems, Yangjiang, China, 12 July - 3 August 2010, World Meteorological Organization Instruments and Observing Methods, Report IOM-107, WMO/TD-No. 1580, available at https://www.wmo.int/pages/prog/www/IMOP/publications-IOM-series.html, 2011. [Last access: 19 August 2016.]

Shibata, T., Vömel, H., Hamdi, S., Kaloka, S., Hasebe, F., Fujiwara, M., and Shiotani, M.: Tropical cirrus clouds near cold point tropopause under ice supersaturated conditions observed by lidar and balloon-borne cryogenic frost point hygrometer, J. Geophys. Res., 112, D03210, doi:10.1029/2006JD007361, 2007.

Shibata, T., Hayashi, M., Naganuma, A., Hara, N., Hara, K., Hasebe, F., Shimizu, K., Komala, N., Inai, Y., Vömel, H., Hamdi, S., Iwasaki, S., Fujiwara, M., Shiotani, M., Ogino, S.-Y., and Nishi, N.: Cirrus cloud appearance in a volcanic aerosol layer around the tropical cold point tropopause over Biak, Indonesia, in January 2011, J. Geophys. Res., 117, D11209, doi:10.1029/2011JD017029, 2012.

Vömel, H., Naebert, T., Dirksen, R., and Sommer, M.: An update on the uncertainties of water vapor measurements using cryogenic frost point hygrometers, Atmos. Meas. Tech., 9, 3755-3768, doi:10.5194/amt-9-3755-2016, 2016.

*Based on a number of statements made throughout the paper, I don't think that the authors have a very strong knowledge of cloud physics. They refer to "dense clouds" as those with concentrations of 2 particles per centimeters cubed, the limit to their sensor's measurement capability. This is only the case for some cirrus clouds but certainly not for any low to mid-level water or mixed phased clouds.*

*Hence, the 2 cm3 coincidence threshold really makes this sensor useful only for cirrus clouds and you don't need a polarization detector to tell you that you are in all ice. This makes the CPS just a cloud detector with no useful information on the number concentration until it get high in the atmosphere.*

We admit that this is one of the major limitations of the CPS1. The priority when we started the development of this instrument was placed in developing a very small-mass, disposable instrument easily combined with a radiosonde (see also above for the characteristics of the radiosonde sounding). However, the CPS can distinguish the cloud phase at least as a layer (i.e. for a group of particles), and the phase of a single particle under particular conditions (please see below for more explanation). Also,

we have provided a simple, partial correction algorithm for the number of count and a conversion algorithm to number concentration.

*In addition, the assumptions that are made about the difference between the polarization signal from water and ice are wrong. They state that the polarization by water droplets is nearly zero but if they do the calculations, they will discover that only at 180 degrees is this true. Water droplets will change the polarization state at angles less than 180 degrees and as a function of size. Hence, unless this bias is taken into account, the detection system as currently set up will classify some water drops as ice. This bias is clearly shown in Figs. 6a and 9a.*

First, the laser device, scattering particle, and the two detectors are placed within the same plane (Figure 1), and the polarization of the incident light is parallel to this plane, as described in Section 2.1. (We will add this information also in the caption of Figure 1 of the revised manuscript.) In this case, the polarization of the scattered light also becomes parallel to the plane regardless the scattering angle. In practice, they are not exactly parallel because of the finite volume of the detection area. In the Mie scattering calculations in Appendix A, we have considered this effect by rotating the Stokes parameter.

Second, in the revised manuscript, we will add the following explanation on the factory calibration process:

The rough-surface particles of 30−40 μm diameter are the spores of the Lycopodium clavatum Linnaeus provided by the Association of Powder Process Industry and Engineering (APPIE), Japan. Other rough-surface particles may also work; the key point is that we use certain particles so that we can calibrate the two detectors after the instrument is fully assembled. The sensitivity of the two detectors are differently adjusted so that the calibration particles give zero DOP on average (using 251 particle data). Typically, detector #2 is about three times more sensitive than detector #1; this is consistent with the fact that the polarization plate used has 34−35 % transmittance.

Third, in Figure R1-1, we show the DOP distributions obtained from the standard spherical particle experiments described in Appendix A. The results for the 5−30 μm particles are quite similar to the flight results for water clouds shown in Figures 4a, 6a, and 9a, namely, being distributed mostly at 0.3−1. For the smallest, 2 μm particles, the distribution is longer tailed to smaller (and negative) DOP values. For the 60 and 100 μm particles, whose $I_{55}$ value was often saturated (as described in Appendix A), the distribution still resembles those for the 5−30 μm particles although it is somewhat longer tailed to smaller (but positive) DOP values. Figure R1-1 and corresponding discussion will be placed at the end of Appendix A of the revised manuscript.

[Figure]

**Figure R1-1. Frequency distributions of the degree of polarization (DOP) in 0.05 bins from the laboratory experiments using the standard spherical particles of (a) 2 µm, (b) 5 µm, (c) 10 µm , (d) 20 µm, (e) 30 µm , (f) 60 µm , and (g) 100 µm diameters. Red, green, and blue curves are for the CPS1 #1, #2, and #3 instruments, respectively. For the 10 µm and 20 µm particles (c and d), two sets of experiments on different days are expressed with solid and dotted curves.**

Finally, in the revised manuscript, we will discuss the relationship between the DOP value and the phase of cloud particle more carefully (by referring to the above Figure R1-1 as well) as follows:

When the DOP value is negative, the particle is ice. When the DOP value is positive but less than ~0.3, the particle is most likely ice. When the DOP value is more than ~0.3, the particle is water in many cases, but there is a chance that it is ice (because DOP can take values between −1 and +1 for ice

particles; as shown in the flight results in the paper); for the final judgement, the DOP statistics of the cloud layer to which the particle is belong and the simultaneous temperature value should also be taken into account (as done in the paper).

*2) Error analysis*

*No paper on measurement technique should be allowed publication without a serious error analysis and error propagation. The number "factor of 2" is given for concentration uncertainty but that is based on a very crude analysis. Given the many factors that affect the flow through the sample tube, the particle velocity can fluctuate much more than that.*

The major factors that affect the flow speed within the detection area are the balloon ascent rate (~5 m s$^{-1}$), additional flow of within about +/−2 m s$^{-1}$ due to the payload pendulum motions (depending on the length of the main string, payload configuration, and altitude), and the potential drag within the duct and around the detection area. Among them, it is considered that the balloon ascent rate determines the order of magnitude. As discussed in Section 2.3, we used hot-wire anemometers, with the uncertainty of +/−1 m s$^{-1}$ (k = 2), to confirm this. In the revised manuscript, we will add a new appendix, Appendix B where we will show the results from a CPS with two hot-wire anemometers launched at Moriya at 17:11:47 LT on 23 November 2012 (see Figure R1-2). The two hot-wire anemometers were placed within a 6-cm long duct (with a similar inner cross section to that of the CPS's air inlet), near the two openings, and this duct was attached at the bottom side of a CPS. Thus, the anemometer #1 is located ~4 cm below the detection area, and the anemometer #2 is 9−10 cm below the detection area. Figure R1-2 compares the balloon ascent rate and the flow speed in the duct measured with two anemometers. From the surface up to ~10 km, the balloon ascent rate is 5−6 m s$^{-1}$, flow speed measured by the anemometer #1 is 4−5 m s$^{-1}$, and flow speed for #2 is 3−4 m s$^{-1}$, with increasing discrepancy between #1 and #2 at higher altitudes. Between 10 and 16 km, the balloon ascent rate is ~5 m s$^{-1}$, flow speed for #1 is ~4 m s$^{-1}$, and flow speed for #2 is 1−2 m s$^{-1}$. It is expected that the actual flow speed for #1 and #2 would not differ because of the same cross section of the air flow. However, the measurement results show that this was not the case. Possible reasons include the horizontal difference in the flow speed within the duct and in the location of the two anemometers (i.e. the anemometer #2 may have been located closer to the duct wall during the flight). Also, these anemometers might have had some directivity dependence, which had not been evaluated in the laboratory experiments. Considering the additional flow of up to 2 m s$^{-1}$ due to the pendulum motions, assumption of a constant flow speed of 5 m s$^{-1}$ with the uncertainty of a factor of ~2 (as the contribution from the flow-speed assumption) is not unreasonable.

[Figure]

**Figure R1-2. Vertical profiles of balloon ascent rate (black) and the flow speed within a duct (with a similar inner cross section to that of the CPS's air inlet) attached to a CPS measured with two hot-wire anemometers, one placed ~4 cm below the detection area (#1, red) and the other ~10 cm below the detection area (#2, blue). Taken at Moriya, Japan, launched at 17:11:47 LT on 23 November 2012. For all three profiles, 0.2-km averages were taken.**

*The suggestion that the transit time can be used to estimate the particle velocity is not valid. A number of things impact the transit time. If the authors would have shown a frequency histogram of transit times, I think it would show values varying by more than factors of three. The reason is that measuring the transit time from threshold to threshold makes the transit time sensitive to particles size.*

In the revised manuscript, we will weaken this suggestion (and point out that selected, high-quality particle signal width data are only useful). The signal width data are primarily used to monitor potential particle overlap in this paper. For the flow speed estimation discussed in Section 2.3, the results from the hot-ware anemometers are primarily referred to; in the revised manuscript, description on the estimation using signal width data will be removed.

*Secondly, if the laser used has a Gaussian intensity profile, then the transit time will depend on where*

*the particle passes through the beam and how far from the center of focus.*

This is true. This is one of the sources of uncertainty in the particle signal width data. However, we do not have quantitative estimates on this at the moment.

*Nothing is said about how particles are constrained to go through the most intense portion of the laser beam. What is the sizing error due to passing through edges or away from the center of focus. Laser beams diverge so the scattering intensity from same sized particles can vary widely.*

Particles are not constrained to go through a particular point of the detection area. The sizing error due to this treatment is shown in the standard particle experiments described in Appendix A. Let us note here that in Table A3 and Figure A1 of the original manuscript, we incorrectly used values of standard error of the mean (i.e. $\sigma/\sqrt{n}$) rather than standard deviation ($\sigma$). We will correct this in the revised manuscript. In Figure R1-3 below, we show the revised Figure A1. The horizontal bars in Figure A1 become much longer (i.e. being multiplied by square root of the number of particles used in each experiment shown in Table A3). The variability in the results, expressed here as the standard deviation, is due to the finite and inhomogeneous detection area and non-constraint particle introduction to it. As seen in this (revised) figure, only a very rough estimate of the particle size is possible with the current CPS1.

[Figure]

**Figure R1-3. Revised Figure A1. (Relationship between $I_{55}$ and the diameters of spherical water droplets (left) and hypothetical spherical ice particles (right) based on Table A2 and the summary column of Table A3. The horizontal bars indicate the standard deviation range obtained from the laboratory experiments, while the vertical bars indicate the minimum and maximum diameters obtained from the Mie scattering calculations.)**

*Is there a polarized filter in front of the laser to insure linear polarized light? If not, polarized laser diodes, particularly inexpensive ones, will have some fraction of the light in a different plane of polarization. This will bias the signal measured by the polarized detector.*

There is no polarized filter in the current version of CPS. The effect of this and other factors (in particular, inhomogeneous laser light intensity within the detection area) results in the DOP distribution (i.e., the one not confined to unity) for spherical and water particles as shown in Figure R1-1 and in the flight results.

*How is the sample volume defined? What is the uncertainty in the sample volume? Why use a sample volume that is at last 100 times larger than necessary and limits the concentration due to coincidence? Has this sample volume been mapped out?*

Figure R1-4 shows the details of the detection area. As in the figure, the cross section of the detection area that the detectors effectively see is estimated as ~1 cm × 1 cm, while its vertical extent is ~0.5 cm. See above for our priority when we started the development of this instrument.

[Figure]

**Figure R1-4. Schematic diagram of the CPS, showing the detection area in solid and dotted red lines.**

*3) Calibration.*

*"Rough particles" are mentioned as calibration for the polarization channel. What are rough particles?*

Please see above.

*Where are the calibration curves with water for I125? That would clearly show that there is a size dependent polarization signal for water droplets.*

Please see above (the results from the standard spherical particle experiments, in Figure R1-1). Yes, there is some dependence on size as described above, but it is not so large in the 2−100 μm range.

*Where is the mapping of the sample volume?*

Please see above.

*4) Polarization references*
*There are now a number of instruments that use measurements of the polarization ratio to differentiate spherical from non-spherical particle yet no mention is made of them. This is a significant oversight. There are a number of IN counters that do this, as well as the CAS-POL and CPSPD that differentiate water droplets from ice crystals looking at backscattered, polarized light.*

As suggested by Referee #3, in Introduction of the revised manuscript, we will cite the aircraft instruments, the Cloud Aerosol Spectrometer with Polarization (CASPOL; e.g. Nichman et al., 2016), the Small Ice Detector mark 2 (SID-2; e.g. Cotton et al., 2010), and the Cloud Particle Spectrometer with Polarization Detection (CPSPD; Baumgardner et al., 2014). We also cite the article by Baumgardner et al. (2012; see above) and the textbook by Wendisch and Brenguier (2013) for recent developments on the aircraft instruments. Please note, however, that our focus is on developing a balloon-borne instrument flown with a radiosonde, and thus, as discussed in the beginning of this reply, there are several, very different technological issues and emphases. We will also point out that in the lidar community, the polarization information has been used for cloud measurements since the 1970s (Schotland et al., 1971; see also Sassen, 1991).

References:

Baumgardner, D., Newton, R., Krämer, M., Meyer, J., Beyer, A., Wendisch, M., Vochezer, P.: The Cloud Particle Spectrometer with Polarization Detection (CPSPD): A next generation open-path cloud probe for distinguishing liquid cloud droplets from ice crystals, Atmos. Res., 142, 2-14, doi:10.1016/j.atmosres.2013.12.010, 2014.

Cotton, R., Osborne, S., Ulanowski, Z., Hirst, E., Kaye, P. H., and Greenaway, R. S.: The ability of the Small Ice Detector (SID-2) to characterize cloud particle and aerosol morphologies obtained during flights of the FAAM BAe-146 research aircraft, J. Atmos. Oceanic Technol., 27 (2), 290–303, doi: 10.1175/2009JTECHA1282.1, 2010.

Nichman, L., Fuchs, C., Järvinen, E., Ignatius, K., Höppel, N. F., Dias, A., Heinritzi, M., Simon, M., Tröstl, J., Wagner, A. C., Wagner, R., Williamson, C., Yan, C., Connolly, P. J., Dorsey, J. R., Duplissy, J., Ehrhart, S., Frege, C., Gordon, H., Hoyle, C. R., Kristensen, T. B., Steiner, G., McPherson Donahue, N., Flagan, R., Gallagher, M. W., Kirkby, J., Möhler, O., Saathoff, H., Schnaiter, M., Stratmann, F., and Tomé, A.: Phase transition observations and discrimination of small cloud

particles by light polarization in expansion chamber experiments, Atmos. Chem. Phys., 16, 3651-3664, doi:10.5194/acp-16-3651-2016, 2016.

Sassen, K.: The polarization lidar technique for cloud research: A review and current assessment, B. Amer. Meteorol. Soc., 72 (12), 1848-1866, doi:10.1175/1520-0477(1991)072<1848:TPLTFC>2.0.CO;2, 1991.

Schotland, R. M., Sassen, K., and Stone, R.: Observations by lidar of linear depolarization ratios for hydrometeors, J. Appl. Meteorol., 10 (5), 1011-1017, doi:10.1175/1520-0450(1971)010<1011:OBLOLD>2.0.CO;2, 1971.

Wendisch, M. and Brenguier, J.-L. (eds): Airborne Measurements for Environmental Research: Methods and Instruments, Wiley-VCH Verlag GmbH & Co. KGaA, Weinheim, Germany, 655 pp., doi:10.1002/9783527653218.2013, 2013.

*In summary, unless the authors can better demonstrate that the CPS will provide any useful cloud property data, this paper should not be accepted for AMT.*

Thank you very much again for your comments and suggestions. Please also see the discussion with the other two referees.

---

## Author Comment (AC3) · 2 Sep 2016

**Reply to Referee #3**

Thank you very much for your reviewing our manuscript and providing us valuable comments and suggestions.

*This paper describes the new balloonborne instrument of cloud particles whose signal data is transmitted with a radiosonde to a ground station. The new sensor, CPS has capabilities to count particles larger than about 2 microns and to distinguish between cloud water droplets and ice crystals. The authors also mention some limitations using the test flight data in several launch sites.*

*However, it is not clear what levels of accuracy and range of uncertainties the new sensor has during the ascent flight. Although it might be the first light-weight costeffective balloonborne sensor over the cloud particle size range with a polarization function, the more quantitative evaluations are required for detecting/measuring variety of atmospheric particles.*

*I am ready to recommend the publication after the authors revise the manuscript taking into account the following comments, especially in terms of more supporting facts and figures which clearly indicate the measurement uncertainties and limitations. In my opinion, the manuscript will require a major revision for publication.*

Please see below for our answers to your questions and concerns.

*General comments:*

*Even if the main purpose of the paper is the demonstration of the new sensor, not quantitative discussions in detail, the basic performance of the sensor under the various atmospheric conditions is necessary since the CPS is a balloonborne sensor, rather than a ground-based instrument.*

We have shown the basic performance of the CPS under various atmospheric conditions, i.e., midlatitude stratus (and cirrus) clouds, midlatitude precipitating clouds including mixed phase clouds, tropical thick cloud layers, and tropical upper tropospheric thin cirrus layers. We have also shown the results from laboratory experiments using various standard spherical particles. In the revised manuscript, we will also show the DOP distributions from the laboratory experiments (please see below for more details).

*The manuscript is well written, however, I did not fully understand the strong points of this paper,*

*compared to the current airborne instruments of cloud particles. There are many other airborne instruments for cloud and aerosol particle measurements with the polarization function: for instance, CASPOL (Baumgardner et al., 2001, 2011; Glen and Brooks, 2013; Nichman et al., 2016), SID (Hirst et al., 2001; Cotton et al., 2010), and CPSPD (Baumgardner et al., 2014). These instruments have a higher sensitivity with more detectors, and those should be referred in the paper at least. Also, a polarization optical particle counter has developed and demonstrated (Kobayashi et al., 2014), although it covers the different measurement size range (0.5 – 10 microns) and is the ground-based instrument. Therefore the comparison in laboratory experiments with other sensor or instrument is strongly recommended to exhibit the basic performance of polarization measurements.*

As discussed in the Reply to Referee #1, the weather balloon with a radiosonde is one of the major platforms for upper air sounding, measuring vertical profiles continuously from the surface up to the middle stratosphere in ~100 minutes. The radiosonde temperature and humidity measurements, including those from balloon-borne special hygrometers, have been well characterized by regular and special intercomparison campaigns (e.g. Nash et al., 2011; Vömel et al., 2016). The typical balloon ascent rate is ~5 m s$^{-1}$, which is much smaller than the speed of aircrafts, easing some of the technical challenges that aircraft instruments may face (e.g. shattering of ice crystals; Baumgardner et al., 2012). On the other hand, balloon instruments need to be disposable, operable with small batteries, and of small mass. These factors may result in technical challenges when developing balloon-borne cloud particle instruments. As described in Introduction, the masses of existing balloon-borne cloud particle instruments range from 1 to 6 kg except for COBALD whose mass is ~500 g. The CPS is a balloon-borne instrument with a small mass (~200 g) which is comparable to that of modern radiosondes, and thus is much more easily combined with a radiosonde to detect the "existence" of cloud particles. This may provide advances in understanding of cloud and water vapour processes because this gives us more opportunities to obtain a set of vertical profiles of temperature, humidity, and (existence of) cloud particles compared to the existing much heavier balloon-borne instruments. A combination of radiosonde temperature and humidity measurements with a ground-based or satellite particle remote-sensing measurements (as discussed in Section 3.4, for example) may be another way to obtain such a data set, but the measurement collocation is always an issue (e.g. Shibata et al., 2007). As a disposable instrument, there is limitation in number concentration measurements, though we propose a simple, partial correction algorithm and a simple algorithm to convert the number of count to number concentration. Furthermore, utilizing the polarization information, the CPS can distinguish between water, ice, and mixed phase cloud layers (i.e. for a group of particles; even for a single particle for particular conditions); more detailed explanation on this is given below.

In the revised manuscript, we will cite the aircraft instruments, the Cloud Aerosol Spectrometer with

Polarization (CASPOL; e.g. Nichman et al., 2016), the Small Ice Detector mark 2 (SID-2; e.g. Cotton et al., 2010), and the Cloud Particle Spectrometer with Polarization Detection (CPSPD; Baumgardner et al., 2014). We also cite the article by Baumgardner et al. (2012) and the textbook by Wendisch and Brenguier (2013) for recent developments on the aircraft instruments. Please note, however, that our focus is on developing a balloon-borne instrument flown with a radiosonde, and thus, as discussed above, there are several, very different technological issues and emphases. We will also point out that in the lidar community, the polarization information has been used for cloud measurements since the 1970s (Schotland et al., 1971; see also Sassen, 1991).

We are reluctant to extend our discussion to aerosol instruments. We would like to limit our discussion to balloon-borne cloud particle instruments, though we will add discussion on some aircraft cloud particle instruments as described above.

We have shown the results of our extensive laboratory experiments using standard spherical particles in Appendix A (and will add the DOP information from these experiments at the end of this appendix; please see below for more details). We are happy to participate in future laboratory and field intercomparison projects. Also, because the CPS is a commercial instrument, anyone who is interested can make independent evaluation by themselves.

References:

Baumgardner, D., Avallone, L., Bansemer, A., Borrmann, S., Brown, P., Bundke, U., Chuang, P. Y., Cziczo, D., Field, P., Gallagher, M., Gayet, J.-F., Heymsfield, A., Korolev, A., Krämer, M., McFarquhar, G., Mertes, S., Möhler, O., Lance, S., Lawson, P., Petters, M. D., Pratt, K., Roberts, G., Rogers, D., Stetzer, O., Stith, J., Strapp, W., Twohy, C., Wendisch, M.: In situ, airborne instrumentation: Addressing and solving measurement problems in ice clouds, Bull. Amer. Meteorol. Soc., 93 (2), ES29–ES34, doi:10.1175/BAMS-D-11-00123.1, 2012.

Baumgardner, D., Newton, R., Krämer, M., Meyer, J., Beyer, A., Wendisch, M., Vochezer, P.: The Cloud Particle Spectrometer with Polarization Detection (CPSPD): A next generation open-path cloud probe for distinguishing liquid cloud droplets from ice crystals, Atmos. Res., 142, 2-14, doi:10.1016/j.atmosres.2013.12.010, 2014.

Cotton, R., Osborne, S., Ulanowski, Z., Hirst, E., Kaye, P. H., and Greenaway, R. S.: The ability of the Small Ice Detector (SID-2) to characterize cloud particle and aerosol morphologies obtained during flights of the FAAM BAe-146 research aircraft, J. Atmos. Oceanic Technol., 27 (2), 290–303, doi: 10.1175/2009JTECHA1282.1, 2010.

Nash, J., Oakley, T., Vömel, H., and Li, W.: WMO Intercomparison of high quality radiosonde observing systems, Yangjiang, China, 12 July - 3 August 2010, World Meteorological Organization

Instruments and Observing Methods, Report IOM-107, WMO/TD-No. 1580, available at https://www.wmo.int/pages/prog/www/IMOP/publications-IOM-series.html, 2011. [Last access: 19 August 2016.]

Nichman, L., Fuchs, C., Järvinen, E., Ignatius, K., Höppel, N. F., Dias, A., Heinritzi, M., Simon, M., Tröstl, J., Wagner, A. C., Wagner, R., Williamson, C., Yan, C., Connolly, P. J., Dorsey, J. R., Duplissy, J., Ehrhart, S., Frege, C., Gordon, H., Hoyle, C. R., Kristensen, T. B., Steiner, G., McPherson Donahue, N., Flagan, R., Gallagher, M. W., Kirkby, J., Möhler, O., Saathoff, H., Schnaiter, M., Stratmann, F., and Tomé, A.: Phase transition observations and discrimination of small cloud particles by light polarization in expansion chamber experiments, Atmos. Chem. Phys., 16, 3651-3664, doi:10.5194/acp-16-3651-2016, 2016.

Sassen, K.: The polarization lidar technique for cloud research: A review and current assessment, B. Amer. Meteorol. Soc., 72 (12), 1848-1866, doi:10.1175/1520-0477(1991)072<1848:TPLTFC>2.0.CO;2, 1991.

Schotland, R. M., Sassen, K., and Stone, R.: Observations by lidar of linear depolarization ratios for hydrometeors, J. Appl. Meteorol., 10 (5), 1011-1017, doi:10.1175/1520-0450(1971)010<1011:OBLOLD>2.0.CO;2, 1971.

Shibata, T., Vömel, H., Hamdi, S., Kaloka, S., Hasebe, F., Fujiwara, M., and Shiotani, M.: Tropical cirrus clouds near cold point tropopause under ice supersaturated conditions observed by lidar and balloon-borne cryogenic frost point hygrometer, J. Geophys. Res., 112, D03210, doi:10.1029/2006JD007361, 2007.

Vömel, H., Naebert, T., Dirksen, R., and Sommer, M.: An update on the uncertainties of water vapor measurements using cryogenic frost point hygrometers, Atmos. Meas. Tech., 9, 3755-3768, doi:10.5194/amt-9-3755-2016, 2016.

Wendisch, M. and Brenguier, J.-L. (eds): Airborne Measurements for Environmental Research: Methods and Instruments, Wiley-VCH Verlag GmbH & Co. KGaA, Weinheim, Germany, 655 pp., doi:10.1002/97835276532182013, 2013.

*If the CPS has an advantage of cloud phase determination in natural clouds, it remains ambiguous only from the frequency distribution of the degree of polarization (DOP). Thinking of the ice initiation from supercooled water clouds, not so many particles are detected as ice since the number of natural ice nuclei is generally scarce in the troposphere. So it is quite difficult to detect the signal of few ice crystals from the DOP frequency distribution. Even in the water clouds, there is certain amount of ranges of DOP, seeing from the Figs. 6a and 9a (I think that Fig. 4a must indicate the signals of aerosols, not water droplets. Please see the specific comment below.).*

First, in the revised manuscript, we will add the following explanation on the factory calibration

process:

The rough-surface particles of 30−40 μm diameter are the spores of the Lycopodium clavatum Linnaeus provided by the Association of Powder Process Industry and Engineering (APPIE), Japan. Other rough-surface particles may also work; the key point is that we use certain particles so that we can calibrate the two detectors after the instrument is fully assembled. The sensitivity of the two detectors are differently adjusted so that the calibration particles give zero DOP on average (using 251 particle data). Typically, detector #2 is about three times more sensitive than detector #1; this is consistent with the fact that the polarization plate used has 34−35 % transmittance.

Second, in Figure R3-1, we show the DOP distributions obtained from the standard spherical particle experiments described in Appendix A. The results for the 5−30 μm particles are quite similar to the flight results for water clouds shown in Figures 4a, 6a, and 9a, namely, being distributed mostly at 0.3−1. For the smallest, 2 μm particles, the distribution is longer tailed to smaller (and negative) DOP values. For the 60 and 100 μm particles, whose $I_{55}$ value was often saturated (as described in Appendix A), the distribution still resembles those for the 5−30 μm particles although it is somewhat longer tailed to smaller (but positive) DOP values. Figure R3-1 and corresponding discussion will be placed at the end of Appendix A of the revised manuscript.

[Figure]

**Figure R3-1. Frequency distributions of the degree of polarization (DOP) in 0.05 bins from the laboratory experiments using the standard spherical particles of (a) 2 µm, (b) 5 µm, (c) 10 µm , (d) 20 µm, (e) 30 µm , (f) 60 µm , and (g) 100 µm diameters. Red, green, and blue curves are for the CPS1 #1, #2, and #3 instruments, respectively. For the 10 µm and 20 µm particles (c and d), two sets of experiments on different days are expressed with solid and dotted curves.**

Third, in the revised manuscript, we will discuss the relationship between the DOP value and the phase of cloud particle more carefully (by referring to the above Figure R3-1 as well) as follows:

When the DOP value is negative, the particle is ice. When the DOP value is positive but less than ~0.3, the particle is most likely ice. When the DOP value is more than ~0.3, the particle is water in many cases, but there is a chance that it is ice (because DOP can take values between −1 and +1 for ice particles; as shown in the flight results in the paper); for the final judgement, the DOP statistics of the

cloud layer to which the particle is belong and the simultaneous temperature value should also be taken into account (as done in the paper).

Finally, as pointed out by the Referee, there was a local dust event for the case shown in Figures 3 and 4. Please see below at your specific comment on this.

*Or if the CPS has an advantage of counting of particles, the number concentrations in clouds, the sampling volume and counting efficiency are more thoroughly evaluated. To evaluate the sampling volume and its uncertainty during the ascent flight is essential to estimate the number concentration of particles. In the paper, only the rough estimate of detection area, ~0.5 cm3, is written in the text. Only the fixed typical ascent rate, ~5 m/s, is used, although it will be variable from the surface to upper levels even in the same case.*

As described in the manuscript, there is an upper limit in the number concentration directly measured by the CPS, though we have proposed a partial correction algorithm. Figure R3-2 shows the details of the detection area. As in the figure, the cross section of the detection area that the detectors effectively see is estimated as ~1 cm × 1 cm, while its vertical extent is ~0.5 cm.

[Figure]

**Figure R3-2**. **Schematic diagram of the CPS, showing the detection area in solid and dotted red lines.**

Instead of using different flow speed values at different data points in a profile, we assume a constant flow speed of 5 m s$^{-1}$ in this paper, and in the revised manuscript we will have a new appendix, Appendix C where we discuss the impact of different flow speeds on the number density conversion and the count number correction (see the next paragraph and Figure R3-3). In this way, the uncertainty in the number concentration values from the CPS is better understood, and the future CPS users can use this information to process and evaluate their data.

Planned Appendix C:

Both the conversion from number of particles $N$ [s$^{-1}$] to number concentration $C$ [cm$^{-3}$] and the correction factor for $N$ depend on the flow speed within the detection area $v$ [m s$^{-1}$]. As the cross section of the detection area is ~1 cm$^2$, $C$ is given by $N/(100v)$. As the vertical extent of the detection area is ~0.5 cm, the correction factor for $N$ (see discussion in Sect. 2.3) is $4 \times (psw/(5/v))^3$ if $psw$ is greater than $5/v$, where $psw$ is particle signal width in ms; if $psw$ is smaller than $5/v$, the correction factor is unity. Figure C1 shows the relationships between $N$ and $C$ and between $psw$ and the correction factor for $v = 3$, 5, and 7 m s$^{-1}$. This indicates the contribution of the uncertainty in $v$ (see also Appendix B) to the uncertainty in (corrected) $N$.

[Figure]

**Figure R3-3. (Planned Figure C1.) Dependence on the flow speed within the detection area ($v$ in m s$^{-1}$) for (a) the conversion from number of particles in s$^{-1}$ to number concentration in cm$^{-3}$ and (b) the correction factor for the number of particles ($N$ in s$^{-1}$) with respect to the particle signal width values in ms. Shown are for 5 m s$^{-1}$ (black), 3 m s$^{-1}$ (blue), and 7 m s$^{-1}$ (red).**

*Is the real counting efficiency considered to be uniform inside the cross section of detection area (1cm x 1cm)? Even if the intensity profile of a laser device and the air flow inside the area are assumed to be uniform, in fact, the efficiency and uncertainty of counting particle might not be distributed with uniformity between in the center and near the edge of the detection area. Does the intensity and peak wavelength of the laser change during the ascent or under different ambient conditions? If so, does it influence the measurement accuracy at what amount of uncertainties?*

The counting efficiency may be roughly estimated by the cross section of the detection area (1 cm$^2$) divided by the cross section of the inlet duct (1.2 cm$^2$), being 83 %. Furthermore, the non-uniformity in the laser devise may result in smaller efficiencies for smaller particles (particularly in the range <30 µm). The temperature dependence of the laser light intensity was evaluated in a laboratory chamber using a CPS and changing the temperature from +25°C to −30°C. Note that during the actual flights (e.g. going through the cold tropical tropopause down to −90°C), the laser device is kept relatively warm thanks to the Styrofoam flight box and heat from the CPS circuit board; for example, during a flight the temperature in the flight box was −25°C when the air temperature was −70°C. The chamber result was that the detector output changed less than 2 % in this temperature range. We have not evaluated the temperature dependence of the laser peak wavelength. In Appendix A, we have taken the possible range (i.e. 775−810 nm) into account. The information in this paragraph will be included in the revised manuscript.

*Moreover, how are the capabilities of measuring coarse aerosols (for instance, dust or sea-salt particles) in real atmosphere? In other words, does the CPS have the capability to distinguish among water droplets, ice crystals, and various types of coarse aerosols over the micron-sized range?*

Any particles greater than 2 µm are detected by the CPS. Please see below for the case in Figures 3 and 4.

*Specific comments:*

*The following comments do not cover all general points above, but explain specifically as well as some minor points.*

*Page 3, Lines 19-20: Why the two photo detectors are placed at angles of 55 and 125 degrees? Do these set of angles have the best performance to distinguish between water droplets and ice crystals?*

As written in the manuscript, we started the CPS development from the PS2 pollen sensor (with the detectors placed at 60 and −60 degrees). As we found that the lower detector for the PS2 type tends to be subject to wet contamination, we decided to move it to a higher place. There were two major considerations to determine the final two angles: (1) the laser device and the two detectors can be placed without difficulty in terms of assembling and production; and (2) the DOP distributions for spherical and non-spherical particles are well (similarly) separated as in the PS2 sensor.

*Page 3, Lines 29-31: As for the calibration or ground test before the launch, in what ways is the sensitivity of both channels confirmed to be equal in terms of size measurements? In what ways is the calibration of polarization signal done quantitatively using the rough-surface particles? And the information on the rough-surface particles is required in the text (commercially available? mean size and its standard deviation? refractive index?).*

As described above, the rough-surface particles used in the factory calibration process are the spores of the Lycopodium clavatum Linnaeus provided by the Association of Powder Process Industry and Engineering (APPIE), Japan. The sensitivity of the two detectors are differently adjusted so that the calibration particles give zero DOP on average (using 251 particle data). Typically, detector #2 is about three times more sensitive than detector #1; this is consistent with the fact that the polarization plate used has 34−35 % transmittance. The particles are commercially available, and of "30−40 μm" diameter stated by the APPIE. Unfortunately, there is no information on the standard deviation of the size and the refractive index.

*Page 4, Lines 1-11: The paper describes the laboratory data focused on the relationship between I55 and the particle diameter. The performance of I125 is also needed to be examined to show the accuracy and uncertainty.*

Thank you very much for your suggestion. As written above, in the revised manuscript we will show the DOP distributions obtained from the laboratory experiments (see Figure R3-1).

*Page 6, Line 5: Using the hot-wire anemometer sensor data, the flow speed in the duct is compared*

*with the balloon ascent rate. Thinking of the outlet area (3cm x 3cm) is wider than the inlet area (2cm x 2cm), what is the mechanism of significantly higher speed in the duct in some cases?*

The uncertainty of the hot-wire anemometers used here is $+/-1$ m s$^{-1}$ (k=2). Furthermore, there may be additional flow of up to $+/-2$ m s$^{-1}$ due to the payload pendulum motions (depending on the length of the main string, payload configuration, and altitude).

*Page 6, Lines 11-18: The actual maximum number counted per second is thought be much smaller than the estimated maximum number (~1000) from limitation of the interface board. What was the maximum number per second measured from the test flight data (in water clouds)? The upper limit of number concentrations in the CPS measurements can be much smaller than ~2 cm-3? The typical number concentration of cloud droplets in natural clouds covers from tens to thousands of particles per cubic centimeter, so basically is it difficult to reliably measure the number concentrations of water clouds even with development of any correction algorithms?*

In water cloud, ~200 s$^{-1}$ for midlatitude precipitating clouds (Figure 5) and ~300 s$^{-1}$ for tropical low level cloud layer (Figure 8) were obtained. They are uncorrected values.

*Page 8, Lines 22-23: The manuscript states that there is a cloud layer from the surface to ~2km. However it is considered more reasonable that the signal should indicate the layer of coarse aerosols (possibly in swollen state) since the RH is subsaturated with respect to water and it is inconceivable to have very low concentrations for cloud water droplets in the subsaturated layer. That is why I think it is quite difficult to determine the cloud phase only from the DOP frequency distribution.*

As described in Page 9, lines 5−8, there was a local dust event on the day of flight (18 March 2013) due to strong surface wind in association with a low pressure system. This is supported by surface observations of enhanced suspended particulate matter (SPM; particles equal to or smaller than 10 μm) at five sites located 3−14 km from the launch site (data obtained from the environmental database of the National Institute for Environmental Studies (NIES)). Sugimoto et al. (2016) also discussed this local dust event based on depolarization-lidar and surface particle (PM$_{2.5}$ and PM$_{10}$) measurements at Tsukuba (at NIES; 36.05°N, 140.12°E), which is located ~16 km from the launch site. These dust particles are mineral dust blown up by the strong wind from unplanted fields in this area. On the other hand, surface precipitation data are available every 10 minutes at Tsukuba (also called as Tateno) site of the Japan Meteorological Agency (36.06°N, 140.13°E; also ~16 km from the launch site). The data record indicates that there was no precipitation at the surface up to 19:50 LT, and after 20:00 LT there was precipitation of less than 0.5 mm. Figure R3-4 shows the results from the depolarization lidar

measurements at Tsukuba (NIES) between 06:00 LT and 24:00 LT on 18 March 2013 (Sugimoto et al., 2008, 2016; http://www-lidar.nies.go.jp/AD-Net/, accessed 15 August 2016). This lidar system uses 532 nm and 1064 nm laser light, with the depolarization measurement capability at 532 nm. It takes 15 minutes for one cycle of the measurement, including 5-min laser-light emission and 10-min temporary pause. The vertical resolution of the data is 30 m. Figure R3-4 shows that the local dust event, confined up to ~0.5 km, started around 09:00 LT and ended around 19:00−20:00 LT. The dust particles have depolarization ratios of ~15−30 %. On the other hand, at 18:00 LT, low-level clouds appeared whose base was estimated to be located at 0.72 km, and the cloud base descended gradually afterwards. These clouds are characterized by large backscatter coefficients (greater than $1 \times 10^{-5}$ m$^{-1}$ sr$^{-1}$) and small depolarization ratios (less than 10%). The two CPSs were launched at Moriya (35.94°N, 140.00°E) at 18:07:44 LT. Therefore, these CPSs should have encountered mineral dust particles (which might have been somewhat wet) up to ~0.5 km and water cloud particles above ~0.7 km.

[Figure]

**Figure R3-4. Time-height distributions of attenuated backscatter coefficient at 532 nm (top) and volume depolarization ratio at 532 nm (bottom) measured with the lidar system at Tsukuba (36.05°N, 140.12°E) between 06:00 LT and 24:00 LT on 18 March 2013. Temporal resolution is 15 minutes, and vertical resolution is 30 m, with the lowest 120 m portion of the data not shown because of large uncertainty. For the depolarization ratio plot, data points whose 532 nm backscatter coefficient is $<1.0 \times 10^{-6}$ m$^{-1}$ sr$^{-1}$ are not shown. Also shown with dots in both panels are the cloud base location estimated from the 1064 nm lidar measurements (i.e. the lowest level where vertical gradient of 1064 nm attenuated backscatter coefficient exceeds $1.2 \times 10^{-6}$ m$^{-1}$ sr$^{-1}$ per 60 m; and the maximum attenuated backscatter coefficient in the cloud layer must exceed $2.0 \times 10^{-5}$ m$^{-1}$ sr$^{-1}$, with apparent cloud top where attenuated backscatter coefficient value becomes equal to that at the cloud base.).**

[Figure]

**Figure R3-5. As for Figure 3 in the originally submitted manuscript, but with time in vertical axis, between -300 s before the launch and +500 s after the launch (time 0 corresponds to the launch time). Results from one of the two CPSs (the CPS1 type) at Moriya, Japan, launched at 18:07:44 LT on 18 March 2013. Time series are shown of (a) temperature (black), RH (blue solid), and ice saturation RH (dashed blue) from the RS-06G radiosondes, (b) number of particles counted per second (and number concentration assuming 5 m s$^{-1}$ flow speed), original in black and corrected in red, (c) particle output voltages $I_{55}$ (red) and $I_{125p}$ (black), (d) the degree of polarization (DOP), (e) particle signal width in ms, and (f) the DC component from the detector #1 output.**

Figure R3-5 shows the results from one of the two CPSs (the CPS1 type) during the period between 300 s before the launch and 500 s after the launch (around 2.5 km). The CPS was placed outside at the surface from ~300 s before the launch. The results show that before the launch (i.e. at the surface), the CPS detected particles whose DOP is distributed widely between −1 and +1 and that after the launch, the DOP became much more confined to ~0.5−0.9. The former group of particles are considered to be the mineral dust particles, and the latter group of particles are mostly spherical particles, i.e. mostly water cloud particles.

In the revised manuscript, we will add the above descriptions and discussions including Figure R3-4. It is true that the CPS would not be able to distinguish between spherical non-water particles (e.g. the standard spherical particles used in the laboratory experiments) and water particles only from the DOP information. Other information such as radiosonde temperature and RH measurements and independent particle measurements is sometimes necessary for this distinction. It should be noted that the uncertainty of the RS-06G and RS-11G RH measurements is +/−7 %RH (this information will also be added in the revised manuscript). Also, water cloud particles can exist temporarily in subsaturated layers when, e.g. they are falling from the above. Finally, we will revise Figure 4 of the originally

submitted manuscript by changing the height region for water clouds from 0.5−1.5 km to 0.8−1.5 km (see Figure R3-6). Essentially, the results do not differ between the original Figure 4(a) and Figure R3-6(a).

Reference:

Sugimoto, N., Shimizu, A., Matsui, I. and Nishikawa, M.: A method for estimating the fraction of mineral dust in particulate matter using $PM_{2.5}$-to-$PM_{10}$ ratios, Particuology, 28, 114–120, doi:10.1016/j.partic.2015.09.005, 2016.

Sugimoto, N., Matsui, I., Shimizu, A., Nishizawa, T., Hara, Y., Xie, C., Uno, I., Yumimoto, K., Wang, Z., Yoon, S.-C.: Lidar network observations of troposheric aerosols, Proc. SPIE 7153, Lidar Remote Sensing for Environmental Monitoring IX, 71530A (December 09, 2008), doi:10.1117/12.806540, 2008.

[Figure]

**Figure R3-6. Revised Figure 4 of the originally submitted manuscript. The height range for (a) has been changed from 0.5−1.5 km to 0.8−1.5 km.**

*Page 11, Lines 21-29: Although it might be impossible to compare the figures related to depolarization between the CPS and the remote-sensing measurements, is there any way to evaluate the DOP uncertainty? The comparison with other airborne sensors in laboratory experiments is useful to show the CPS performance for depolarization.*

The DOP uncertainty can be evaluated by the laboratory experiments using standard spherical particles. Please see above (and Figure R3-1). It is impossible to relate the CPS's DOP values to the lidar depolarization ratio values. We are happy to participate in future laboratory and field intercomparison projects.

*Pages 13-14 (Appendix A): The laboratory experiments are described using the standard spherical particles with different types and sizes. The description on how to enter the particles into the detection area is also required in the text. Is the air vacuumed from the bottom of the particle inlet duct at 5 m/s? How do the particles dispense before entering the particle inlet? Are there any calibration data to be described using non-spherical particles? That might be helpful to show the uncertainty of the CPS polarization measurement.*

The air was not vacuumed. We used a special apparatus to introduce the particles separately with a fan that pushes air downward into the CPS. Non-spherical particles were not used in the laboratory experiments, but as described above, the rough-surface particles are used during the factory calibration process.

Thank you very much again for your comments and suggestions. Please also see the discussion with the other two referees.

---

## Author Comment (AC2)

**Reply to Referee #2 (Dr. F. Wienhold)**

Thank you very much for your reviewing our manuscript and providing us valuable comments and suggestions.

*This contribution presents a new lightweight balloon-borne Cloud Particle Sensor (CPS). It describes the instrumental setup with the constraints imposed by the conditions on operational soundings balloons, and explains how particle concentrations is estimated from the primary count rate. Results from test flights are presented that compare two instrumental setups and provide simultaneous measurements with the Cryogenic Frostpoint Hygrometer (CFH). Various midlatitude and tropical cloud layers are probed illustrating the deployment in a wide range of cloud types. This clearly demonstrates that different cloud types (pure water, mixed phase to cirrus) can be well identified and characterized.*

*The paper is well organized. The current development state of the new sensor and the experience gained in various balloon sounding deployments deserve publication in Atmospheric Measurement Techniques. However, limitations and uncertainties need to be addressed more clearly, and a roadmap for future development should be added.*

Thank you very much for your understanding and positive evaluation. Please see below for our answers to your questions.

*General comments*

*The term degree of polarization (DOP) is introduced unconventionally including differently adjusted detector sensitivities leading to an observed range from -1 and 1, whereas in common notion it would be expected to range from 0 to 1. To avoid confusion this needs to be stated more explicitly – see specific comments.*

In the revised manuscript, we will add the following explanation on the factory calibration process:
The rough-surface particles of 30−40 μm diameter are the spores of the Lycopodium clavatum Linnaeus provided by the Association of Powder Process Industry and Engineering (APPIE), Japan. Other rough-surface particles may also work; the key point is that we use certain particles so that we can calibrate the two detectors after the instrument is fully assembled. The sensitivity of the two detectors are differently adjusted so that the calibration particles give zero DOP on average (using 251 particle data). Typically, detector #2 is about three times more sensitive than detector #1; this is

consistent with the fact that the polarization plate used has 34−35 % transmittance.

*The air flow speed in the instrument is essential both to convert the count rate into number density, and to correct for signal overlap caused by multiple particle detection. Substantial flow speed reduction with respect to the ascent rate is reported for the instrument geometry used, rough agreement is found only for larger cross sections tested. Equating the flow speed with the typical sounding balloon ascent rate of 5 m/s – with a resulting reference signal width of 1 ms – is inappropriate when the effects number density conversion and overlap correction are not discussed (see also specific comments). Low number density regions near ground in Figures 5 and 7 with signal widths in excess of 1 ms conflict the undiminished flow assumption. The corrected particle count rate should be clearly addressed as an upper bound estimate, not just as corrected value.*

First, in the revised manuscript, we will add a new appendix, Appendix B where we will show the results from a CPS with two hot-wire anemometers (with the uncertainty of +/−1 m s$^{-1}$ (k = 2)) launched at Moriya at 17:11:47 LT on 23 November 2012 (see Figure R1-2). The two hot-wire anemometers were placed within a 6-cm long duct (with a similar inner cross section to that of the CPS's air inlet), near the two openings, and this duct was attached at the bottom side of a CPS. Thus, the anemometer #1 is located ~4 cm below the detection area, and the anemometer #2 is 9−10 cm below the detection area. Figure R1-2 compares the balloon ascent rate and the flow speed in the duct measured with two anemometers. From the surface up to ~10 km, the balloon ascent rate is 5−6 m s$^{-1}$, flow speed measured by the anemometer #1 is 4−5 m s$^{-1}$, and flow speed for #2 is 3−4 m s$^{-1}$, with increasing discrepancy between #1 and #2 at higher altitudes. Between 10 and 16 km, the balloon ascent rate is ~5 m s$^{-1}$, flow speed for #1 is ~4 m s$^{-1}$, and flow speed for #2 is 1−2 m s$^{-1}$. It is expected that the actual flow speed for #1 and #2 would not differ because of the same cross section of the air flow. However, the measurement results show that this was not the case. Possible reasons include the horizontal difference in the flow speed within the duct and in the location of the two anemometers (i.e. the anemometer #2 may have been located closer to the duct wall during the flight). Also, these anemometers might have had some directivity dependence, which had not been evaluated in the laboratory experiments. Considering the additional flow of up to 2 m s$^{-1}$ due to the pendulum motions, assumption of a constant flow speed of 5 m s$^{-1}$ with the uncertainty of a factor of ~2 (as the contribution from the flow-speed assumption) is not unreasonable.

[Figure]

**Figure R2-1. Vertical profiles of balloon ascent rate (black) and the flow speed within a duct (with a similar inner cross section to that of the CPS's air inlet) attached to a CPS measured with two hot-wire anemometers, one placed ~4 cm below the detection area (#1, red) and the other ~10 cm below the detection area (#2, blue). Taken at Moriya, Japan, launched at 17:11:47 LT on 23 November 2012. For all three profiles, 0.2-km averages were taken.**

Second, in the revised manuscript we will have a new appendix, Appendix C where we discuss the impact of different flow speeds on the number density conversion and the count number correction (see the next paragraph and Figure R2-2). In this way, the uncertainty in the number concentration values from the CPS is better understood, and the future CPS users can use this information to process and evaluate their data.

Planned Appendix C:

Both the conversion from number of particles $N$ [s$^{-1}$] to number concentration $C$ [cm$^{-3}$] and the correction factor for $N$ depend on the flow speed within the detection area $v$ [m s$^{-1}$]. As the cross section of the detection area is ~1 cm$^2$, $C$ is given by $N/(100v)$. As the vertical extent of the detection area is ~0.5 cm, the correction factor for $N$ (see discussion in Sect. 2.3) is $4 \times (psw/(5/v))^3$ if $psw$ is greater than $5/v$, where $psw$ is particle signal width in ms; if $psw$ is smaller than $5/v$, the correction factor is unity. Figure C1 shows the relationships between $N$ and $C$ and between $psw$ and the correction

factor for $v = 3$, 5, and 7 m s$^{-1}$. This indicates the contribution of the uncertainty in $v$ (see also Appendix B) to the uncertainty in (corrected) $N$.

[Figure]

**Figure R2-2. (Planned Figure C1.) Dependence on the flow speed within the detection area ($v$ in m s$^{-1}$) for (a) the conversion from number of particles in s$^{-1}$ to number concentration in cm$^{-3}$ and (b) the correction factor for the number of particles ($N$ in s$^{-1}$) with respect to the particle signal width values in ms. Shown are for 5 m s$^{-1}$ (black), 3 m s$^{-1}$ (blue), and 7 m s$^{-1}$ (red).**

Finally, are the "Low number density regions near ground in Figures 5 and 7 with signal widths in excess of 1 ms" the surface to ~500 m region where the signal width is sometimes ~1.5 ms for Figure 5 and ~1.1 ms or ~3 ms for Figure 7? Yes, this indicates that the assumption of 5 m s$^{-1}$ is sometimes not valid. We will more clearly write that the corrected particle count rate should be regarded as an upper bound estimate, not just as corrected value.

*The Meisei radiosonde interface is limited to 25 bytes/s while other radiosonde offer much larger bandwidth as 80 bytes/s for the Intermet Systems iMet or 100 bytes/s for Vaisala RS41. These would*

*allow addressing the above problem by a better characterization of the signal width distribution instead of just transmitting the first six samples. I suggest adding a perspective for future development which could include real-time (onboard) data processing options in conjunction with increased downlink capacity. This could illustrate how the promising potential of this sensor could be exploited further.*

Thank you very much for your suggestions. We assume that both iMet and RS41 radiosondes support the XDATA protocol. The authors from Meisei will consider to work with other researchers to develop a processing and interfacing board for these and other radiosondes.

*Specific comments*

*Page 1, line 22: before "of the instrument", replace "volume of the detection area" by "ascent rate together with the detection volume and exposed cross section".*

The number 2 cm$^{-3}$ is a multiplicative inverse of the volume of the detection area (0.5 cm$^3$). If there is more than one particle in the detection area, a count loss would occur.

*Page 3, lines 3-13: A very approximate cost might be indicated here.*

For the first commercial version, the price is about 4−5 times of that of the modern radiosondes.

*Page 3, line 12: The situations covered by the four test flights could be mentioned here.*

We will do so.

*Page 3, lines 16 to 27: The orientation of the rectangular slits should be indicated explicitly in the text or the related Figure 1.*

We will revise the relevant sentence as:
The slit in front of the light source is 0.55 cm (parallel to Figure 1, i.e. the space shown in Figure 1) × 1.0 cm (perpendicular to Figure 1), while the slits in front of the two detectors are 0.50 cm (parallel) × 1.0 cm (perpendicular).
Please also see Figure R1-4 in the Reply to Referee #1.

*Pages 3, line 30: do the "rough-surface particles" cause full depolarization and thus about equal sensitivity for both channels, or is the cross-polarized detector tuned more sensitive due to calibrating with only partially depolarized light?*

Yes for both. Please see the answer to the first general comment.

*Page 4, lines 6, 7: replace "gives" by "determines", and "other factors" by "polarization".*

Will be replaced.

*Page 4, line 15 (compare comment p 3, l 30): To not confuse the reader it should be stated that for equally sensitive detector the DOP should be between 0 and 1. As detector #1 senses the sum of co- and cross-polarized scattered light, negative DOP values can only result from higher sensitivity of detector #2.*

Thank you very much for this suggestion. Please see the answer to the first general comment.

*Page 4, line 16: mentioning "the uncertainty in the factory calibration": Is there clear calibration target?*

Please see the answer to the first general comment.

*Page 4 line 20: Replace "various" by "nonzero".*

"Various" is correct. Please see the answer to the first general comment.

*Page 4, lines 26, 27: It should be stated that particle overlap excludes flow rate determination: the two signal-width interpretations are complementary.*

Yes, we will add this point.

*Page 5, line 12: Replace "is 1.0 cm longer" by "extends 1.0 cm".*

Will be replaced.

*Page 5, line 19: Other radiosondes offering external instrument interfaces provide higher bandwidth, e. g. 80 bytes/s (Intermet iMet-1RS) or 100 bytes/s (Vaisala RS41). This additional capacity could be used in the future development perspective.*

We will add this point as a possibility.

*Page 6, line 10: The experimental findings described in this section confirm that the flow rate through the instrument is reduced, as expected. With the tested larger cross section the flow agrees with the ascent rate, but for the configuration used in what follows it is only about half the balloon ascent rate. It should be stated that assuming the nominal flow speed of 5 m/s implies under-estimating both the count rate and the 1 ms reference signal width.*

Please see the answer to the second general comment. Note that if the actual flow speed was e.g. 3 m s$^{-1}$, the threshold signal width should be 1.67 ms; thus, assuming 5 m s$^{-1}$ for this case would result in overestimation of the correction factor *f* and the corrected count rate.

*Page 6, line 27: The signal width factor f enters the count rate correction to the third power. An assessment of the consequences should be provided.*

As explained in the answer to the second general comment, we will include the information on the quantitative consequences of this treatment.

Page 7, line 5: Replace "is close or even greater" by "exceeds", and delete "highly" in line 6: Ice saturation between 120% and 130% are frequently observed without cloud formation.

Will be replaced.

*Page 8, line 10: Description of future work could be anchored here which allows for an improved signal width characterization rather than just sending the first six samples.*

We will describe this possibility here and in Section 4.

*Page 8 line 11: Not calculating the DOP for signals exceeding 7 V contradicts the statement on page 4, lines 17 to 22.*

We did not calculate the DOP for signals exceeding 7 V because a strong peak appears at zero DOP

for "ice" particle cases, while the main message on page 4, lines 17 to 22 was that larger "spherical" (and thus water) particles might still give a DOP distribution for spherical particles (please also see Figure R1-1 in the Reply to Referee #1). In practice, all the flight cases shown in the paper did not detect "water" particles with signals exceeding 7 V (Figs. 3c, 3i, 5c, (7c), and 8c), while they detected "ice" particles with signals exceeding 7 V. Therefore, there is no contradiction. However, we will add more explanation here so that the readers would not be confused.

*Page 9, line 24: What is the major difference of "the first commercial version" flown 2015 versus the instrument launched 2013? Near ground level signal width of Figure 5 appears to have increased with respect to Figure 3.*

The major difference is in the Styrofoam flight box. In and before 2014, it was made by hand. Also, the length of the inlet duct is slightly different. Thus, there should be no difference in quality in the signal width data. The difference between Figures 3 and 5 near the ground is considered to be due to the actual difference in the nature of the particles. For the case of Figure 3, please see also the discussion with Referee #3.

*Page 10, line 3: Excluding DOP analysis again contradicts page 4, lines 17 to 22.*

Please see above. We will add more explanation so that the readers would not be confused.

*Page 10, lines 10-11 and 31: The authors set a (debatable) signal width limit of 1 ms as onset for particle overlap correction (see p 6, l 24). This should be reflected using 1 instead of 0 as lower limit in the ranges given for the signal width.*

For all the current figures, we set the lower limit as −1.0, to clearly show all the data points. As in the answer to the second general comment (Figure R2-2), we will address the issue of the choice of signal width threshold in a different way.

*Page 10 line 33: "for some reason" needs further specification: instrumental or telemetry failure?*

Probably, this is related to the CPS board or the interface board, but there also is a possibility of telemetry failure.

*Page 11, line 15: See signal width range comment for page 10.*

Please see above.

*Page 11, lines 16-20: Supersaturated layers above cirrus cloud tops are not surprising and have been discussed in detail in Brabec et al. (2012), including microphysical modeling.*

Thank you for pointing to the paper by Brabec et al. for the supersaturation issue. We will cite this paper again here.

*Page 11, line 21: Which of the lidar wavelengths mentioned is used for analysis and displayed in Figure 10?*

The backscattering coefficient is at 1064 nm, and the depolarization ratio is at 532 nm. We will add this information in the revised manuscript.

Thank you very much again for your comments and suggestions. Please also see the discussion with the other two referees.